# XTransfer: Modality-Agnostic Few-Shot Model Transfer for Human Sensing at the Edge

**Yu Zhang** [1]  **Xi Zhang** [1]  **Hualin Zhou** [1]  **Xinyuan Chen** [1]  **Shang Gao** [1]  **Hong Jia** [2]  **Jianfei Yang** [3]  **Yuankai Qi** [1]  **Tao Gu** [1]

## Abstract

Deep learning for human sensing on edge systems presents significant potential for smart applications. However, its training and development are hindered by the limited availability of sensor data and resource constraints of edge systems. While transferring pre-trained models to different sensing applications is promising, existing methods often require extensive sensor data and computational resources, resulting in high costs and limited transferability. In this paper, we propose `XTransfer`, a first-of-its-kind method enabling modality-agnostic, few-shot model transfer with resource-efficient design. `XTransfer` flexibly uses pre-trained models and transfers knowledge across different modalities by (i) *model repairing* that safely mitigates modality shift by adapting pre-trained layers with only few sensor data, and (ii) *layer recombining* that efficiently searches and recombines layers of interest from source models in a layer-wise manner to restructure models. We benchmark various baselines across diverse human sensing datasets spanning different modalities. The results show that `XTransfer` achieves state-of-the-art performance while significantly reducing the costs of sensor data collection, model training, and edge deployment.

## 1. Introduction

Human sensing refers to the process of capturing and interpreting data related to human activities, behaviors, and physiological states using various sensors (Zhang et al., 2023a). With the proliferation of edge devices equipped with sensors, human sensing plays a vital role in edge systems to understand contexts and enable smart applications, ranging from activity recognition to emotion or vital sign detection (Ahamed & Farid, 2018). Deep learning (DL) offers robust performance in interpreting sensor data, and its deployment on edge devices improves privacy and reduces bandwidth (Zhang et al., 2020; Chen et al., 2020). However, existing DL solutions are often data- and resource-intensive for training and deployment (Dhar et al., 2021), posing significant challenges for data collection and edge deployment in human sensing (Sobin, 2020; Liu et al., 2019).

Unlike other data modalities (*e.g.*, vision or text), collecting human sensing data for training can be costly and even impractical. This is uniquely due to the data that presents brittleness (*i.e.*, noise sensitivity, low SNR), inconsistency (*i.e.*, user variance, scenario changes), and heterogeneity (*i.e.*, differences in hardware configuration) (Lane & Georgiev, 2015; Jeyakumar et al., 2019; Teh et al., 2020). Collecting such data typically requires ethical approvals, as it involves sensitive information (*e.g.*, location, motion patterns) that raises privacy concerns (Zhang et al., 2022). Manual annotation of large volumes of sensor data is labor-intensive and incurs high costs (Song et al., 2023). Importantly, human sensing spans various modalities (*e.g.*, IMU, ultrasound, mmWave radar) and data types (*e.g.*, spectrograms, Doppler profiles, time series), further amplifying the costs of applying DL solutions and naturally leading to cross-modality scenarios (Xiao et al., 2022; Zhang et al., 2023a).

Prior works on Few-shot Learning (FSL) (Wang et al., 2024), transfer learning (Dhekane & Ploetz, 2024), and cross-domain FSL (Thukral et al., 2025; Yin et al., 2024) reduce data collection costs by adapting pre-trained models to sensing applications. However, they typically rely on large-scale labeled datasets within the same modality to costly train source models from scratch (Song et al., 2023), or face challenges such as *modality shift* with limited transferability when leveraging pre-trained models (Oh et al., 2022; Tao et al., 2022; Guo et al., 2022), particularly in cross-modality settings. Recent advances in multi-modal learning (Yang et al., 2023; Chen & Yang, 2025; Chen et al., 2024a; Madaan et al., 2024; Yang et al., 2024; Baldenweg et al.,

[1]Macquarie University, Sydney, NSW, Australia [2]The University of Auckland, Auckland, New Zealand [3]Nanyang Technological University, Singapore. Correspondence to: Yu Zhang <y.zhang@mq.edu.au>, Tao Gu <tao.gu@mq.edu.au>.

*Proceedings of the 43rd International Conference on Machine Learning*, Seoul, South Korea. PMLR 306, 2026. Copyright 2026 by the author(s).

2024) (*e.g.*, CLIP-based models (Girdhar et al., 2023; Wu et al., 2023; Moon et al., 2023)) and cross-modal learning (*e.g.*, image-to-sensor distillation (Zhao et al., 2018; Song et al., 2022; Gurbuz et al., 2020; Zhang et al., 2023c) or alignment-based transfer (Kamboj et al., 2025; Chen et al., 2024b)), further demonstrate the feasibility of knowledge transfer across modalities. However, they remain modality-specific (*i.e.*, relying on shared semantic feature spaces), and typically require large-scale unlabeled and paired data with resource-intensive training and deployment, leading to high costs and making them impractical for few-shot adaptation to support human sensing at the edge. These gaps motivate a *scalable* and *adaptable* model transfer that can broadly reuse pre-trained models across modalities (*e.g.*, vision or text) for new sensing modalities (*e.g.*, radar or bio-signals) using only few data.

In practice, however, addressing this need is uniquely challenging. Due to substantial differences in data characteristics, such as shape, distribution, and feature representation, between source and target modalities (*i.e.*, *modality shift*), transferred models often suffer from significantly degraded performance (*i.e.*, *negative transfer*) (Guo et al., 2022; Oh et al., 2022). Our experiments reveal that state-of-the-art (SOTA) baselines (Table 1) fail to reach oracle performance (Gong et al., 2019) and suffer from substantial *overfitting* when applied to human sensing tasks with few data, falling notably short of the *goal*, as shown in Figure 1(a). These highlight the underlying challenge of *latent feature distribution misalignment* across modalities. Also, these methods often neglect *resource constraints*, rendering them impractical for edge deployment, especially in advanced multi-source settings (Zhao et al., 2020; Yue et al., 2021a; Lee et al., 2019). While model structuring methods such as neural architecture search (NAS) (Wen et al., 2023; Han et al., 2021), pruning (Frankle & Carbin, 2019; Shen et al., 2022), or quantization (Polino et al., 2018), aim to reduce resource demands, they often fail to maintain performance under modality shift and few data. As shown in Figure 1(c), our experiments using pruning (Shen et al., 2022) demonstrate significant model accuracy loss by 50.5% on average.

Driven by the increasing availability of high-quality pre-trained models from public repositories (*e.g.*, PyTorch Hub (PyTorch, 2026)), we propose XTransfer, a first-of-its-kind method enabling modality-agnostic, few-shot model transfer. It flexibly leverages public pre-trained models and transfers knowledge across modalities (*e.g.*, image, text, or audio) to human sensing tasks using only few labeled sensor data with no additional unlabeled data (*i.e.*, *cross-modality FSL settings*, Appendix A.1), significantly reducing the costs of large-scale data collection and training from scratch. Its resource-efficient design also optimizes both cloud performance and edge deployment, enabling a scalable and adaptable method across human sensing applications.

Instead of relying on shared semantic spaces or paired cross-modal data, XTransfer is motivated by the intuition that even when a source model is not explicitly aligned with the target modality, its pre-trained discriminative latent representations may still contain reusable cross-modal structure if target features can be properly aligned. This intuition is also supported by recent work (Gupta et al., 2026), which shows that useful cross-modal structure can be leveraged even without explicit paired cross-modal data. Building on this, XTransfer mitigates modality shift by *model repairing* at the level of *layer-wise* latent feature distribution misalignment using activation-based statistics. This layer-wise formulation is motivated by our observation (Section 3) that modality shift "damages" intermediate representations unevenly across layers, making global adaptation with few data unstable and prone to overfitting. It is enabled by a Splice–Repair–Removal (SRR) pipeline (Section 4) that (i) splices heterogeneous layer inputs, (ii) repairs layer misalignment via an anchor-based generative transfer module in a reduced-orthogonal feature space, and (iii) removes unnecessary channels after repair. Building on the insight that not all repaired layers contribute, XTransfer further features *layer recombining* using a Layer-Wise Search (LWS) control (Section 5) that selects and recombines *layers of interest* (*i.e.*, useful repaired layers) through an NAS-inspired, resource-constrained layer-wise search, incrementally enhancing the repairing and optimizing model structure. To *accelerate* and *stabilize* LWS in multi-source settings with large candidate pools, we incorporate a *pre-search check* strategy coupled with a *dynamic search range* mechanism to avoid unnecessary repairing, balancing accuracy and efficiency. The main contributions are summarized as:

- A first-of-its-kind method, XTransfer, that enables modality-agnostic, few-shot model transfer with resource-efficient design for human sensing on edge systems.
- A SRR pipeline that mitigates modality shift and prevents overfitting via layer-wise repairing of latent feature distribution misalignment.
- A LWS control that enables efficient, stable layer-wise search, selectively recombining layers of interest while discarding others for model restructuring.
- Extensive experimental evaluations (Section 6) under cross-modality FSL settings, benchmarking SOTA baselines (Table 1) on various source and target datasets (Table 2), showing consistent accuracy and efficiency gains.

## 2. Related Work

**Learning with few data for human sensing.** Recent works in FSL and cross-domain FSL have proposed *single-* and

---

[1]We test a pre-trained ResNet18 on miniImageNet as source and HHAR dataset as target under a 5-shot setting with LOOCV enabled (details in Appendix A.2).

[2]The number of classes in the source model should be greater than that in the target.

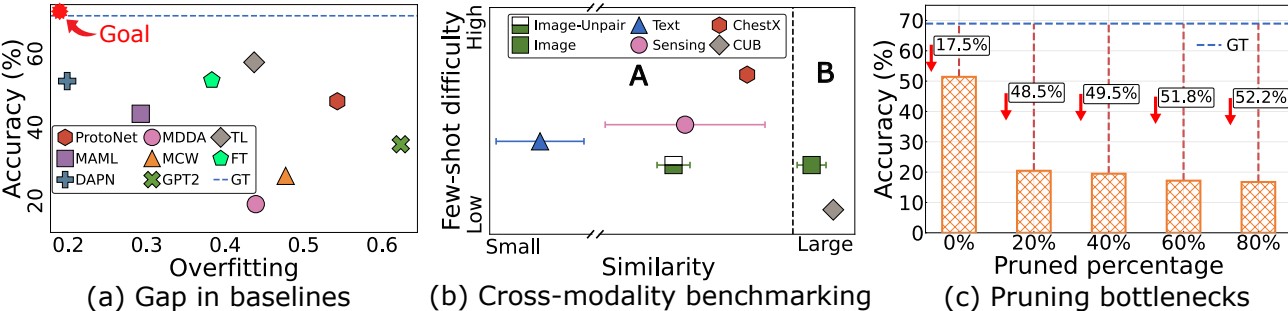

*Figure 1.* Preliminary study under cross-modality FSL settings [1]. **(a)** reveals baseline performance gap. **(b)** shows average similarity and FSL difficulty across all target sensing datasets in Table 2 to source modalities (*e.g.*, Image, Text, Sensing) using default reshaping (Appendix B.1) and benchmarking (Oh et al., 2022) (Appendix A.2 for details). Two distinct areas represent similarity levels (A–hard, B–normal). Key findings: 1) compared to CUB, similarity levels across modalities are notably low, *e.g.*, Text and Sensing fall into Area A, indicating a significant modality shift; 2) compared to Image-Unpair (*i.e.*, no class pairing) in Area A, Image surprisingly falls into Area B, indicating that pairing classes [2] may enhance cross-modality similarity; 3) Image exhibits more stable standard deviations and lower FSL difficulty, suggesting better potential for model transfer. **(c)** shows significant model accuracy loss using pruning.

*multi-source* methods. Single-source methods align latent features between source models and target data via distance-based objectives (Zhao et al., 2021; Yue et al., 2021b), whereas multi-source methods leverage the Wisdom of the Crowd principle through unsupervised source distillation (Zhao et al., 2020; Yue et al., 2021a) or maximal correlation analysis (Lee et al., 2019). Recent efforts have also explored fine-tuning LLMs (*e.g.*, GPT2 (Zhou et al., 2023)) for time-series analysis. However, testing on benchmarks (Liang et al., 2021; Oh et al., 2022), our results reveal that they suffer from severe overfitting under cross-modality FSL settings. LLM-based data augmentation (Leng et al., 2024; 2023) helps reduce data collection costs, but remains modality-specific and orthogonal to our focus. Recent advances introduce foundation models (FM) (Abbaspourazad et al., 2024; Weng et al., 2024) or prompt-based FM adaptation (Li et al., 2025) for human sensing, but these methods assume either a pre-existing target-modality FM or access to large-scale unlabeled and paired data for feature alignment. Existing attempts in human sensing FSL (Gong et al., 2019; Wang et al., 2024; Thukral et al., 2025; Yin et al., 2024) remain constrained by a strong reliance on learning in the same modality, resulting in costly source data collection and training. In contrast, XTransfer enables layer-wise model repairing to safely mitigate modality shifts using only few sensor data, achieving significant cost reduction.

**Model structuring for edge deployment.** Model structuring techniques such as pruning (Frankle & Carbin, 2019; Shen et al., 2022), quantization using low-precision data types (Polino et al., 2018), model merging by fine-tuning shared layers (Padmanabhan et al., 2023), one-size-fits-all model (Cai et al., 2020), and NAS techniques such as model restructuring (Wen et al., 2023; Han et al., 2021) or search using few data (Eustratiadis et al., 2024; Xu et al., 2022), have shown success in reducing model resource overhead. However, they strongly rely on either sufficient tar-

get datasets or minimal modality shifts to maintain model performance. XTransfer uniquely applies layer recombining to optimize model structures while strengthening re-paired layer-wise dependence at scale under cross-modality FSL settings. It not only restructures models for streamlined edge deployment but also enhances model repairing.

## 3. Preliminary

**Layer-wise analysis.** It is widely used to diagnose representation quality and guide model restructuring (Sahoo et al., 2025). We examine how accuracy and latent feature discriminability change across layers. Since effective training relies on discriminative latent representations (Islam et al., 2021), we quantify layer-wise latent features using Mean Magnitude of Channels (MMC) (Luo et al., 2022) as a lightweight activation-based statistic. We also measure class clustering via the Silhouette score (S-score) (Shahapure & Nicholas, 2020) computed on MMC-based representations, which captures inter-/intra-class distances and reflects layer-wise discriminability. We segment the backbone into *L-units* (*i.e.*, source layers), defined as a single layer or an inseparable dependent layer block (L-Blocks in Table 3) that cannot be structurally decoupled (*e.g.*, residual blocks in ResNet).

The experimental setup is detailed in Appendix A.3, and additional evidence on S-score/accuracy correlation is provided in Appendix A.4. Figure 2(a) shows that negative transfer (Zhang et al., 2023b) disrupts accuracy convergence across layers (*i.e.*, impaired layer-wise dependence), leading to misalignment and accuracy drops under few-shot fine-tuning. These degradations correlate with MMC shifts, where larger shifts correspond to larger accuracy drops (Figure 2(b)). We adopt MMC as a modality-agnostic metric as it is defined purely on layer activations without assuming shared semantic spaces or paired data, remaining lightweight and stable under FSL settings, compared to

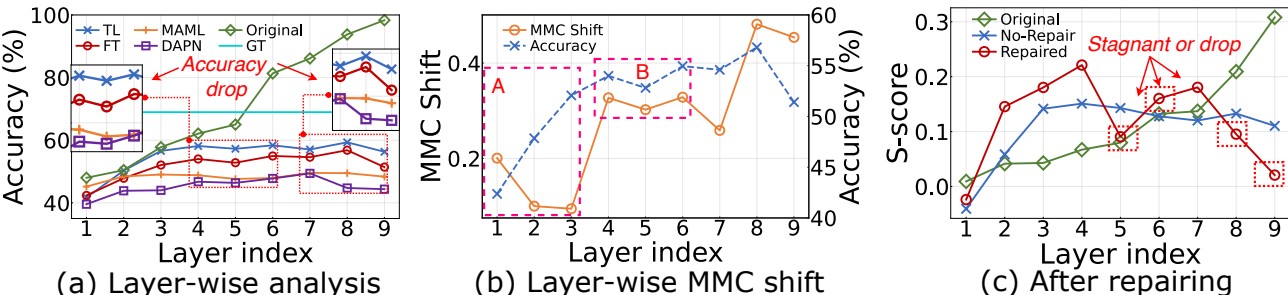

*Figure 2.* Design insights. **(a)** Layer-wise accuracy convergence using baselines is disrupted due to modality shift. **(b)** In area A, accuracy rises as MMC shift stays low, indicating a small latent feature gap. In area B, MMC shift notably increases with layer index, where excessive latent feature deviation begins to reduce accuracy. **(c)** After repairing, S-score improves but stagnation occurs at certain layers.

kernel-based distribution metrics such as Maximum Mean Discrepancy (MMD) (Wang et al., 2023).

**Problem setup.** We aim to minimize *layer-wise MMC shifts* under cross-modality FSL settings. This poses several challenges that motivate the design of SRR pipeline and LWS control: (i) cross-modality transfer must handle structural differences (*e.g.*, input formats and feature shapes) and semantic discrepancies (*e.g.*, non-overlapping label spaces), requiring an objective defined purely on latent representations (*e.g.*, activation-based MMC statistics) rather than shared semantic spaces or paired data; (ii) the latent feature distribution misalignment is complicated by *noisy* representations and strong layer-wise dependencies inherited from pre-training, making few-shot repair brittle; and (iii) not all layers remain useful after repairing (Figure 2(c)), selectively recombining useful layers is critical and must be done efficiently and stably in multi-source settings with large candidate pools. Figure 3 overviews XTransfer.

## 4. Model Repairing

Enabled by SRR pipeline, this process safely repairs layer-wise misalignment by minimizing MMC shifts, restoring pre-trained layer performance under cross-modality FSL settings. As shown in Figure 3, SRR comprises three stages. **Splice:** instead of default reshaping using fixed up/down-sampling (Appendix B.1), which is non-trainable and can discard features, especially when layer shapes vary during layer-wise search, we propose a trainable, compact *connector* placed between heterogeneous layers to enforce shape compatibility, consisting of a Pre-header (adaptive convolutional layer), Resizer, and one encoder-decoder pair. **Repair:** each connector is fine-tuned by the *generative transfer module* to adapt layer channels and minimize MMC shifts. The repair is target-specific, where the corresponding connectors are initialized and repaired for each target modality. **Removal:** *layer channel removal* is applied to further strengthen repairing.

### 4.1. Repairing in complex feature space

Since source layers are already highly discriminative, our intuition is to *safely* adapt few sensor data by aligning their misaligned latent feature distributions to the "original" source-layer distributions, which serve as *anchors*. We hence propose cross-modality *anchor-based* alignment, which aligns source MMCs as *anchors* (*i.e.*, anchor MMCs) with sensing MMCs. Since the raw MMC feature space is high-dimensional, noisy and correlated across channels, our insight is to reduce MMC dimensionality by preserving key layer channels while suppressing noise.

**Reduced-orthogonal feature space.** For each layer, we project anchor MMCs into a low-dimensional, orthogonal feature space via PCA (Valpola, 2015) (*i.e.*, *anchor PCA space*). PCA removes redundancy by decorrelating channel dimensions and suppresses noise, yielding a compact subspace (we use 2 components by default) that preserves maximal variance and typically forms highly clustered class distributions. The resulting projection coefficients (*i.e.*, *component weights*) also highlight the most important layer channels that contribute to layer performance. Given high discriminability of the original layers, the projected anchor MMCs preserve high clustering performance, with class distributions tightly centered (*i.e.*, Anchor-center), as shown in Figure 3(a). Once initialized per layer, the same anchor PCA space is reused to project sensing MMCs.

**Feature space alignment.** After projection, we observe that the sample distribution for each class, along with their centroid (*i.e.*, Sensing-center), appears disorganized compared to the well-clustered Anchor-center at each layer, as shown in Figure 3(a). Due to MMC shifts, the *scale* and *orientation* of Sensing-center are notably different from those of Anchor-center. To align the scale, we define a scale function based on the mean shift in inter-class distance $\text{InterD}(\cdot)$:

$$S = \frac{\text{Mean}(\text{InterD}(\text{Pro}(fs_{ij})))}{\text{Mean}(\text{InterD}(\text{Pro}(ft_{ij})))} \tag{1}$$

where $\text{Pro}(\cdot) = \text{PCA}(\text{MMC}(\cdot))$ denotes the space project function, $fs$ and $ft$ represent the latent features from both

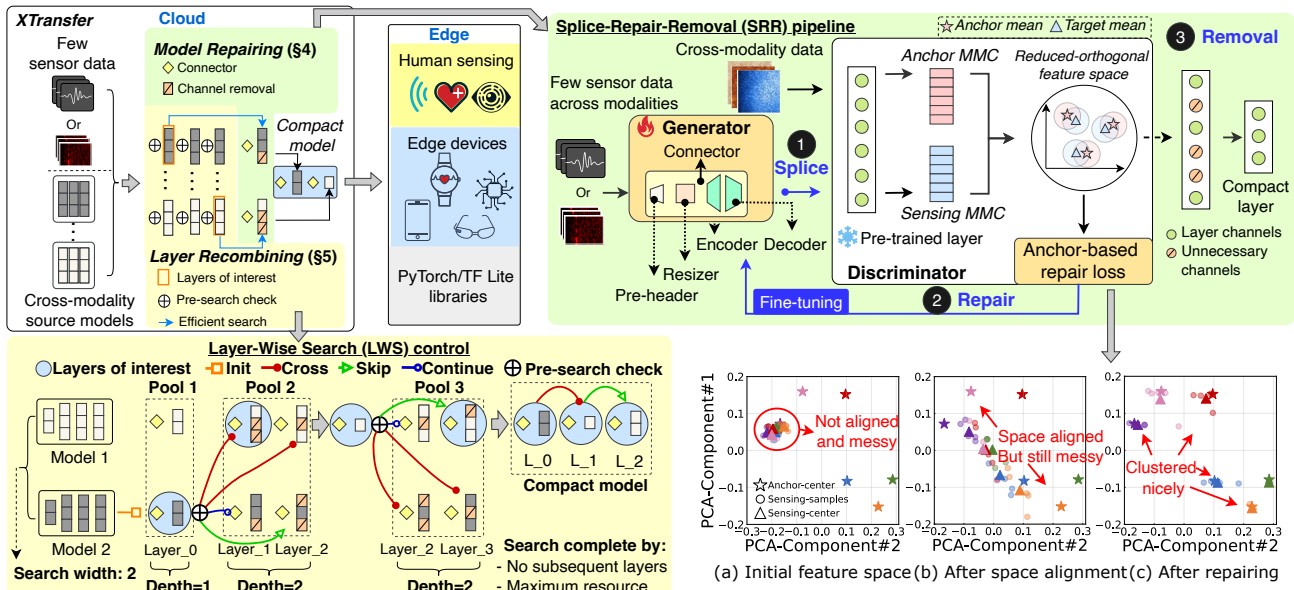

*Figure 3.* Overview. `XTransfer` transfers source models across modalities with few sensor data through model repairing (SRR pipeline) and layer recombining (LWS control). LWS control first segments source models into layers and operates layer-wise search across pools. At each pool, the pre-search check decides which layers need repairing, then SRR pipeline repairs them and LWS control selects layers of interest. These layers are incrementally recombined during the search, restructuring models for enabling human sensing at the edge. Subfigures (a)–(c) illustrate the feature space evolution before and after repairing.

source and sensing modalities, respectively. $i$ denotes the index of pre-trained models, and $j$ denotes the layer index in each model $i$. Next, to align the orientation, we design a rotation alignment function that minimizes the cosine angle (Nguyen & Bai, 2010) by fine-tuning a multi-dimensional rotation matrix $\mathbf{M}_{\text{rot}}$ with dimensions matching the number of PCA components:

$$\underset{\mathbf{M}_{\text{rot}}}{\arg\min} \operatorname{Cos}(\operatorname{Cen}(\operatorname{Pro}(fs_{ij})), \operatorname{Cen}(\operatorname{Pro}(ft_{ij})\,S)\mathbf{M}_{\text{rot}}) \quad (2)$$

where $\operatorname{Cos}(\cdot)$ denotes the cosine similarity function, and $\operatorname{Cen}(\cdot)$ computes the centroid of the projected distribution, defined as the mean vector of the projected samples at each layer. Figure 3(b) shows the aligned results.

**Anchor class pairing.** Building on the key finding that class pairing boosts latent feature alignment across modalities in Figure 1(b), we propose selecting source classes (*i.e.*, the original label classes of the source models) with the highest S-scores as *anchor* classes to pair with sensing classes (*i.e.*, one-to-one source–target class matching) by minimizing a pairing shift objective (*i.e.*, sum of centroid distances across pairs). This is solved as a linear sum assignment via the standard Hungarian algorithm, outputting the optimal pairing set $\mathcal{P}_{\text{ST}}$. This pairing is fully optimized by similarity in the given feature space.

### 4.2. Repairing layer-wise dependence

**Generative transfer module.** To restore layer-wise dependence, we design a *generative transfer module* that opti-

mizes the generator (*i.e.*, connector) using an adversarial objective. The discriminator operates on each frozen pre-trained layer together with its anchor PCA space (Figure 3). Freezing preserves stable anchors and prevents overfitting. Given each pairing set $\mathcal{P}_{\text{ST}}$, the connector is fine-tuned to minimize MMC shifts, uniquely transforming sensor inputs to align with anchors.

**Anchor-based repair loss.** We propose an *anchor-based repair loss* ($loss_{srr}$) to harmonize layer-wise latent feature distributions across modalities in the anchor PCA space. For each class pair $(c_s, c_t) \in \mathcal{P}_{\text{ST}}$, our objective (i) *minimizes* the Euclidean distance $D(\cdot)$ between the centroids of the projected anchor and sensing MMC distributions (*i.e.*, positive loss), and (ii) *maximizes* the distance between sensing MMC samples with different labels ($c_t^-$) (*i.e.*, negative loss). Since the projected anchor MMCs are highly clustered, they can provide the maximum *margin* ($M_{max}$) between anchor clusters. To further improve the negative loss, we update the function by calculating the *anchor-based margin* $margin^c = \operatorname{InterD}(\operatorname{Pro}(fs_{ij}^c)) - \operatorname{IntraD}(\operatorname{Pro}(fs_{ij}^c))$, where the $M_{max}$ is selected, and $\operatorname{IntraD}(\cdot)$ measures the intra-class distance. The loss function is formulated as:

$$loss_{srr} = \frac{1}{N} \sum_{(c_s, c_t) \in \mathcal{P}_{\text{ST}}}^{N} D(\operatorname{Cen}(\operatorname{Pro}(fs_{ij}^{c_s})), \operatorname{Cen}(\operatorname{Pro}(ft_{ij}^{c_t})))$$
$$+ \frac{1}{K} \sum_{(c_t, c_{tk}^-)}^{K} \operatorname{ReLU}(M_{max} - D(\operatorname{Pro}(ft_{ij}^{c_t}), \operatorname{Pro}(ft_{ij}^{c_{tk}^-}))) \quad (3)$$

where $K$ denotes the number of negative samples, and $N$

denotes the number of classes. The $\text{ReLU}(\cdot)$ is used to trigger the negative loss in the condition where the distance between the samples is less than $M_{max}$. Figure 3(c) shows the effectiveness of model repairing.

### 4.3. PCA-based layer channel removal

Layer channel removal continues to enhance layer-wise performance after repairing by removing unnecessary channels at each layer. Existing pruning techniques (Shen et al., 2022; Frankle & Carbin, 2019) typically rely on channel-importance metrics such as the L2 norm computed from MMC. However, under MMC shift, such metrics may inaccurately estimate channel importance and thus degrade performance. To address this, we propose using PCA-based layer channel importance, derived from the optimized component weights during repairing, as a more reliable metric. Our objective is to remove unnecessary layer channels $\widetilde{ch}$ while preserving the maximum S-score $\text{Score}(\cdot)$:

$$\underset{\widetilde{ch} \in CH}{\arg\max} \, \text{Score}\Big(\text{PCA}\Big(\text{MMC}(ft_{ij}^r), \widetilde{ch}\Big), Y_t\Big), \qquad (4)$$

where $CH$ denotes the set of all layer channels, $ft_{ij}^r$ denotes the repaired target latent features, and $Y_t$ denotes the few target labels.

## 5. Layer Recombining

Working with the SRR pipeline, XTransfer performs *layer recombining* via LWS control to (i) perform the pre-search check that accelerates the search, (ii) stabilize the search using the dynamic search range mechanism, and (iii) select and recombine layers of interest to restructure models.

### 5.1. Search control design

Inspired by NAS (White et al., 2023), effective layer-wise search relies on properly designing the search space, actions, and strategy under on-device resource constraints. We hence define the search space as all layers from source models, organized into *search pools* containing layer candidates (*i.e.*, repaired layers). Each pool forms a window of size $I \times J$, where $I$ is the number of source models (width) and $J$ is the number of L-units (depth). We also define four search actions: *init* selects a starting layer (default depth 1); *continue* and *skip* perform intra-model recombination; and *cross* enables inter-model recombination.

**Resource-constrained search strategy.** To effectively control search actions within each search pool, we propose a layer value function based on the S-score $\text{Score}(\cdot)$ to evaluate the performance of each repaired layer and the convergence between adjacent layers after each SRR process:

$$V_{ij} = \text{Score}(\text{SRR}(\text{Pro}(ft_{ij}), MT_{ij}, L_{ij}), Y_t), \qquad (5)$$

where $MT_{ij}$ is the connector and $L_{ij}$ is the repaired layer.

Under resource constraints, LWS control selects the highest-valued candidate in each pool and verifies that it improves layer-wise dependence by checking whether its S-score exceeds that of the previously selected layer. If selected, $L_{ij}$ becomes a *layer of interest* and is recombined, and the next pool starts from layer index $j + 1$. Otherwise, the pool is discarded, and the window shifts forward. To incorporate the constraints, we define a resource coefficient $\text{RC}(n) = \exp(n/(L-2) - 2) + 1$, where $n \in [1, L]$ is the current recombined layer index and $L$ is the maximum number of L-units. Each candidate's adjusted overhead is computed as $R(n)_{ij} = \text{Res}(L_{ij}) \cdot \text{RC}(n)/\max \text{Res}(P)$, where $\text{Res}(\cdot)$ measures actual cost (*e.g.*, weighted FLOPs and memory), and $\max \text{Res}(P)$ is the maximum cost in the current pool $P$. The resource-constrained layer value becomes $VR_{ij} = V_{ij}/R(n)_{ij}$. The objective is $\arg\max_{L_{ij} \in P} VR_{ij}$, subject to $\sum_{m=1}^{n} \text{Res}(L_m) \leq R_{\max}$, where $R_{\max}$ is the device-specific resource budget, and $L_m$ denotes the $m$-th selected layer in the recombined model.

### 5.2. Efficient search

**Pre-search check.** Although SRR fine-tunes connectors efficiently, applying SRR to every candidate in a large pool can still slow down LWS control. We hence propose the *pre-search check* strategy to avoid unnecessary repairs *before* applying SRR. We track the S-score progression across three states: 1) after anchor pairing (Anchor) (*i.e.*, repairing objective), 2) before SRR (Before), and 3) after SRR (After). Empirically, SRR improves S-scores from Before to After, approaching the Anchor level, which indicates a strong correlation (see Appendix B.2 for details). Based on this insight, our intuition is to estimate repaired S-score (*i.e.*, repair rate) of each layer from the Before and Anchor states to decide whether unnecessary repairs can be bypassed. Experiments show that the actual repair rate grows exponentially with the recombined layer index (Appendix B.3). We hence design a *repair rate growth model* as $rate_n^{est} = \exp(an) + b$, where $n \in [1, L]$, $a$ and $b$ are optimized via non-linear regression by minimizing

$$\underset{a,b}{\arg\min} \, \frac{1}{n'} \sum_{m=1}^{n'} (rate_m^{est} - rate_m)^2 \qquad (6)$$

where $n'$ is the number of recombined layers used for fitting and $m$ is the layer index in the recombined model. Estimation starts after *init* and is updated after each layer selection.

**Dynamic search range mechanism.** Using $rate_n^{est}$, LWS can initially apply a narrow search range (*e.g.*, only the top-1 candidate by estimated repair rate) to filter most candidates *before* SRR. However, since $rate_n^{est}$ is updated sparsely, estimation errors may cause the optimal candidate to be missed. To stabilize the search while retaining efficiency, we develop a dynamic search range mechanism that adjusts the search range as $range_n = (1 + \hat{u}^{\pm} rate_n^{est}) \, range_{n-1}$,

where $range_n \in [0.2, 1]$. The range is initialized to 0.5 and updated based on the previous range. Here, $\hat{u}$ is a signed unit vector. If $rate_{n-1}^{est} < rate_{n-1}$, this indicates that the estimation may be inaccurate, and $\hat{u}$ becomes positive to enlarge the search range to repair more candidates and stabilize accuracy. Otherwise, the search range is reduced to bypass unnecessary repairs and improve efficiency.

## 6. Experiment

### 6.1. Experiment setup

We benchmark XTransfer [1] against SOTA baselines, grouped into single- and multi-source methods (Table 1). Evaluation is conducted on 8 widely-used public datasets under cross-modality FSL settings (Table 2), using their corresponding public pre-trained models as sources, from diverse modalities (*e.g.*, image, text, audio, and sensing) with various backbones such as ResNet18 from PyTorch Hub (PyTorch, 2026) (Table 3). To evaluate XTransfer, we develop 4 real-world applications and build testbeds on 3 commercial edge devices (*e.g.*, smartwatch, smartphone, Raspberry Pi, Appendix Table 6 for details).

Based on the testbeds, we collect 4 human sensing datasets (marked as Private in Table 2) as targets, from 40 participants in real-world settings, with ethics approval from the Human Research Ethics Committee of our institute. Additionally, we include 2 public sensing datasets and 1 image dataset as targets for standard benchmarking. We apply standard Leave-One-Out Cross-Validation (LOOCV) (Gong et al., 2019) for evaluating human sensing datasets, and adopt accuracy-to-resource (ATR) ratio, defined as $ATR = Accuracy/(\alpha \, \text{Norm}(\text{FLOPs}) + (1-\alpha) \, \text{Norm}(\text{Params}))$ for evaluating resource-accuracy efficiency, where $\alpha$ is set to 0.5 by default (see Appendix A for more details on our experiment setup).

### 6.2. XTransfer performance

**Repair performance.** We first evaluate how our Repair loss reduces MMC shift to enhance each layer's S-score with different MMC dimensionalities. We use HHAR and the standard pre-trained ResNet18 on miniImageNet under 5-shot setting. The layer channel removal is disabled. We test with 4 different levels of MMC dimensionality (*i.e.*, number of layer channels) ranging from 64 to 512. We employ N-Pair loss (Zhang et al., 2023c) and Triplet loss (Weller et al., 2022) in FSL sensing tasks as baselines. We run 15 rounds for each setting by default throughout the evaluation. Figure 4(a) presents that our Repair loss consistently achieves the highest S-score of 0.2 on average, outperforming both N-Pair loss (0.05) and Triplet loss (0.07), across all

---

levels of MMC dimensionality. It indicates that Repair loss effectively enhances layer performance.

**Efficient search performance.** We now evaluate LWS control performance regarding model accuracy and layer search efficiency. We employ HHAR as target, ResNet18 on miniImageNet and Multi-ResNet18 on 5 image datasets as sources shown in Table 2, with a default *search depth* of 3 in 3-, 5- and 10-shot settings. We also utilize MetaSense in Table 1 as the oracle baseline. We compare different configurations: ResNet18 (Single) and Multi-ResNet18 (Multi) without efficient search enabled, Multi-ResNet18 (Multi-Pre) with only pre-search check using top-1 layer selection enabled, and Multi-ResNet18 (Multi-Efficient) with both pre-search check and dynamic search range enabled. As shown in Figure 4(b), Multi-Pre achieves the lowest search time on average across all n-shot settings, but it leads to degraded accuracy by 3.36% and 7.94% on average, respectively, compared to Multi and Multi-Efficient. It also fails to reach oracle-level accuracy and presents the highest standard deviation on average. The results indicate that relying solely on the pre-search check introduces search stability issues due to inaccurate or drifted repair value estimation. In contrast, Multi-Efficient provides a strong balance between accuracy and efficiency. It reduces search time by 2.1 to 4 times compared to Multi in 5- and 10-shot settings, while outperforming the oracle baseline. While Multi-Efficient takes a longer search time than both Single and Multi-Pre, it successfully achieves superior accuracy and search stability, highlighting the benefits of using multiple source models and dynamic search range mechanism. Notably, even the search space is expanded 5 times more compared to Single, Multi-Efficient requires only 2.1 times more search time, indicating strong scalability enabled by the efficient search. The results also reveal that both Multi-Efficient and Multi achieve lower standard deviation on average compared to Single, suggesting that using multiple source models offers commendable search stability. In short, the results prove that the proposed efficient search significantly accelerates the layer search process while improving accuracy and stability.

**Impact of different sources.** Since the selection of pre-trained source models plays a crucial role in XTransfer, we evaluate how different source models from various modalities affect the accuracy and resource overhead of the output models. We use HHAR, 2 pre-trained ResNet18s on both miniImageNet (Image) and VoxCeleb (Audio), and 2 pre-trained ResNet18-1Ds on both Newsgroup (Text) and OPPORTUNITY (Sensing), as shown in Table 2. We keep the other experimental setups unchanged and include 2-shot settings. Figure 4(c) shows that the output models using Image source achieve the highest accuracy of 70.5% on average in 3- to 10-shot settings, compared to that of 69.6%, 64.3%, and 67% achieved by using Text, Audio, Sensing sources, respectively. This suggests that the latent features

---

[1]The code is available at https://github.com/zhangy10/XTransfer.

*Table 1.* State-of-the-art (SOTA) baselines.

| Baselines | Group | Description |
|---|---|---|
| Transfer Learning (TL) (Jiang et al., 2022) | Single-source | A standard transfer learning method that reuses a source model by **freezing all layers**. |
| Fine-tuning (FT) (Oh et al., 2022) | Single-source | A conventional fine-tuning method that **updates all layers** of a pre-trained model using available target data. |
| ProtoNet (Chen et al., 2019) | Single-source | A distance-based FSL method that constructs **class prototypes** by averaging support set and classifies query set in the feature space. |
| MAML (Chen et al., 2019) | Single-source | A gradient-based meta-learning method that learns a source model with **shared initialization parameters** from scratch. |
| DAPN (Zhao et al., 2021) | Single-source | A cross-domain FSL method that uses **adversarial ProtoNet** with a domain discriminator to reduce latent feature distribution shift. |
| SemiCMT (Chen et al., 2024b) | Single-source | A cross-modal learning method that applies **contrastive self-supervised learning** during model adaptation. |
| GPT2 (Zhou et al., 2023) | Single-(LLM) | A LLM-based method that enables **in-context learning** after fine-tuning on few target data. |
| MDDA (Zhao et al., 2020) | Multi-source | A multi-domain FSL method that uses **adversarial discriminative adaptation** and **multi-source knowledge distillation**. |
| MCW (Lee et al., 2019) | Multi-source | A multi-domain FSL method that uses **maximal correlation analysis** to fuse multiple source domains for few-shot model adaptation. |
| MetaSense (Gong et al., 2019) | Oracle-(GT) | An advanced FSL method based on MAML that uses **extensive target data** to train from scratch, serving as an oracle upper bound. |

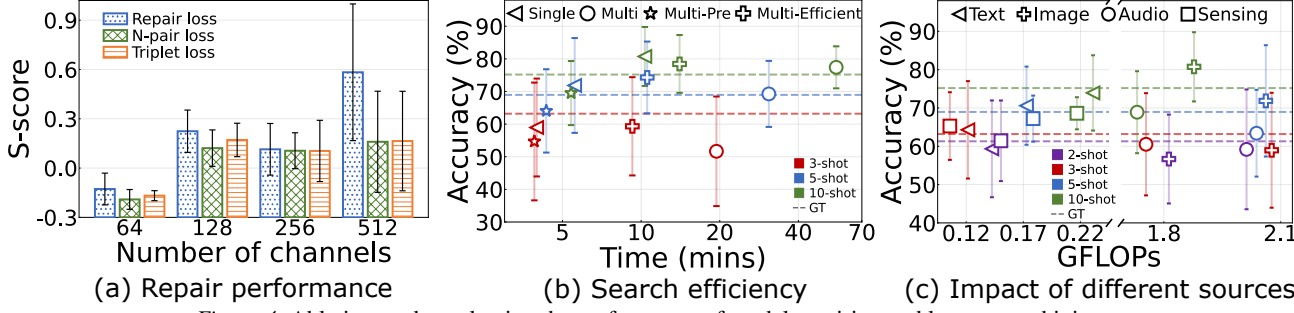

*Figure 4.* Ablation study evaluating the performance of model repairing and layer recombining.

*Table 2.* Source and target datasets specifications.

| Dataset | Modality | Subject | Class | Train Size | Type | Shape | Privacy |
|---|---|---|---|---|---|---|---|
| **Source** | | | | | | | |
| miniImageNet (Vinyals et al., 2016) | Image | - | 100 | 50k | - | 3x84x84 | Public |
| miniDomainNet (Zhou et al., 2021) | Image | - | 126 | 137.5k | - | 3x224x224 | Public |
| Office-31 (Saenko et al., 2010) | Image | - | 31 | 3.3k | - | 3x224x224 | Public |
| Office-Home (Venkateswara et al., 2017) | Image | - | 65 | 12.7k | - | 3x224x224 | Public |
| Caltech-101 (Fei-Fei et al., 2004) | Image | - | 101 | 7.3k | - | 3x224x224 | Public |
| Newsgroup (Thulasidasan et al., 2021) | Text | - | 20 | 15k | - | 100x1x1000 | Public |
| VoxCeleb (Nagrani et al., 2017) | Audio | - | 100 | 14.9k | Time-S | 1x512x300 | Public |
| OPPORTUNITY (Roggen et al., 2010) | IMU | 12 | 11 | 3.7k | Time-S | 77x1x100 | Public |
| **Target** | | | | | | | |
| HHAR (Stisen et al., 2015) | IMU | 9 | 6 | - | Time-S | 6x1x256 | Public |
| WESAD (Schmidt et al., 2018) | ECG | 15 | 3 | - | Time-S | 10x1x256 | Public |
| Gesture | IMU/PPG | 10 | 8 | - | Time-S | 7x1x200 | Private |
| Writing | Ultrasound | 10 | 10 | - | Time-S | 1x1x256 | Private |
| Emotion | mmWave | 10 | 7 | - | Spectrum | 3x224x224 | Private |
| BP | mmWave | 10 | - | - | Doppler | 1x1x499 | Private |
| ChestX (Wang et al., 2017) | Image | - | 5 | - | - | 1x1024x1024 | Public |

*Table 3.* DL model backbone specifications.

| Backbone | Layers | L-Blocks | GFLOPs | Params | Kernel | Shape |
|---|---|---|---|---|---|---|
| ResNet18 (Chen et al., 2019) | 18 | 9 | 3.67 | 11.18M | 2D | 3x224x224 |
| ResNet18-1D (Chen et al., 2019) | 18 | 9 | 0.39 | 3.9M | 1D | 100x1x1000 |
| Multi-Conv4 (Zhao et al., 2020) | 4x5 | 4 | 0.3 | 1.15M | 2D | 3x32x32 |
| Multi-ResNet10 (Zhao et al., 2020) | 10x5 | 5 | 0.7 | 24.5M | 2D | 3x84x84 |
| Multi-ResNet18 (Zhao et al., 2020) | 18x5 | 9 | 18.35 | 55.9M | 2D | 3x224x224 |
| Conv4 (Chen et al., 2019) | 4 | 4 | 0.02 | 0.07 | 1D | 1x1x500 |
| Conv5 (Gong et al., 2019) | 5 | 5 | 0.19 | 30.42M | 1D | By Input |
| GPT2 (Zhou et al., 2023) | 4x6 | 6 | 0.41 | 82.9M | 1D | By Input |

learned from Image, trained on larger datasets with broader class semantics, provide higher discriminability than those learned from other modalities. Notably, Image achieves lower average accuracy in 2- and 3-shot settings, suggesting that extremely low target data may not fully represent the test data distribution, leading to less stable alignment when using Image source. In contrast, both Text and Sensing result in higher average accuracy and reach the oracle, indicating that performance varies with source model quality and semantic relevance to the target. These results also show that XTransfer can flexibly reuse diverse source models across modalities to improve ultra low-shot perfor-

mance. Moreover, the output models using Text or Sensing sources require only 0.16G FLOPs on average, as 1D kernels require much less computation than 2D kernels. Since XTransfer prioritizes accuracy, we select Image source models by default. More ablation results see Appendix C.

### 6.3. Cross-modality FSL performance

We evaluate the overall performance of XTransfer regarding both accuracy and ATR across 5 sensing datasets and 1 image dataset as targets (Table 2). We compare XTransfer against a range of SOTA baselines shown in Table 1. We report two variants with different sources: *Our-Single*, using a ResNet18 pre-trained on miniImageNet, and *Our-Multi*, using Multi-ResNet18 with five corresponding image datasets. We further decouple SRR and LWS by evaluating SRR alone (*i.e.*, *SRR*) and SRR without layer channel removal (*i.e.*, *SRR-w/o-Removal*). We additionally report GPT2-small fine-tuning for comparison, following the parameter setting in (Zhou et al., 2023). All methods on target datasets are evaluated under the same cross-modality FSL settings (Appendix A.1), where only few target data are fine-tuned with sources. MetaSense (Gong et al., 2019) serves as the oracle, trained on Conv5 with sufficient labeled target sensor data (Table 3). For consistency, we also train the oracle using ResNet18 on Emotion to match 2D kernels.

As shown in Tables 4 and 5, both our variants outperform the baselines regarding model accuracy and ATR, and reach or surpass the oracle in accuracy across all target datasets in 5- and 10-shot settings. Specifically, both *Our-Single* and *Our-Multi* achieve a significantly higher ATR on average, surpassing the existing single- and multi-source baselines

| | HHAR (%) | | | WESAD (%) | | | Gesture (%) | | | Writing (%) | | | Emotion (%) | | | ChestX (%) | | |
|---|---|---|---|---|---|---|---|---|---|---|---|---|---|---|---|---|---|---|
| **Method** | 3-shot | 5-shot | 10-shot | 3-shot | 5-shot | 10-shot | 3-shot | 5-shot | 10-shot | 3-shot | 5-shot | 10-shot | 3-shot | 5-shot | 10-shot | 3-shot | 5-shot | 10-shot |
| MetaSense (Gong et al., 2019) | 63.2 | 69.0 | 75.2 | 55.1 | 64.0 | 68.3 | 66.3 | 73.4 | 79.4 | 72.1 | 83.3 | 89.6 | 55.5 | 56.3 | 65.5 | 24.8 | 28.1 | 31.7 |
| ProtoNet (Chen et al., 2019) | 50.6 | 45.7 | 49.8 | 43.9 | 49.3 | 52.0 | **50.8** | 55.8 | 59.2 | 69.5 | 78.7 | 73.4 | **51.3** | 49.2 | 54.8 | 21.7 | 23.4 | 24.5 |
| DAPN (Zhao et al., 2021) | 48.7 | 51.2 | 54.7 | 56.9 | 61.2 | 63.1 | 45.8 | 49.4 | 50.5 | **75.1** | 78.6 | 79.6 | **48.0** | 50.0 | **58.7** | 24.0 | 24.4 | 25.5 |
| MAML (Chen et al., 2019) | 36.6 | 42.3 | 42.4 | 39.6 | 57.9 | 65.6 | 37.2 | 41.3 | 39.4 | 31.6 | 39.7 | 35.2 | 22.3 | 26.9 | 31.1 | 22.5 | 20.4 | 21.9 |
| SemiCMT (Chen et al., 2024b) | 33.3 | 38.9 | 48.9 | 37.3 | 40.0 | 44.4 | 23.5 | 34.4 | 42.5 | 27.3 | 38.6 | 48.0 | 24.0 | 33.6 | 46.7 | 25.3 | 25.6 | 28.0 |
| GPT2 (Zhou et al., 2023) | 41.0 | 34.0 | 45.0 | 48.0 | 35.0 | 36.0 | 49.0 | 54.0 | 63.0 | 52.0 | 71.0 | 75.0 | - | - | - | - | - | - |
| SRR-w/o-Removal | 58.0 | 63.6 | 71.2 | 60.4 | 63.1 | 64.4 | 48.2 | 50.2 | 66.3 | 65.7 | 77.9 | 87.5 | 35.2 | 43.2 | 52.8 | 22.4 | 24.5 | 26.4 |
| SRR | 53.7 | 61.8 | 71.6 | 62.2 | 62.7 | 65.3 | **53.0** | 54.5 | 67.2 | 57.6 | 76.0 | 81.5 | 38.1 | 42.6 | 53.2 | 24.4 | 25.3 | 25.3 |
| Our-Single | **59.0** | **71.8** | **80.7** | **77.9** | **78.4** | **81.8** | 45.2 | **69.6** | **77.8** | 72.8 | **87.0** | **91.3** | 43.6 | **55.6** | **61.4** | **27.7** | **28.6** | **30.4** |
| MDDA (Zhao et al., 2020) | 13.2 | 17.7 | 18.8 | 40.0 | 39.3 | 41.9 | 11.2 | 13.9 | 15.2 | 12.5 | 10.9 | 9.0 | 14.4 | 15.3 | 16.0 | 20.6 | 19.6 | 19.9 |
| MCW (Lee et al., 2019) | 27.6 | 25.4 | 26.9 | 30.7 | 32.7 | 32.4 | 22.3 | 23.4 | 25.6 | 32.3 | 31.6 | 30.2 | 25.3 | 27.2 | 26.5 | 22.4 | 21.5 | 22.8 |
| Our-Multi | **59.3** | **74.3** | **78.4** | **76.4** | **77.8** | **78.7** | 48.7 | **73.1** | **78.6** | 74.1 | **86.1** | **90.4** | 42.3 | **55.1** | 58.6 | **29.0** | **30.0** | **30.2** |

*Table 4.* Comparison in model accuracy. Top-2 results are highlighted in **bold**.

| | HHAR | | | WESAD | | | Gesture | | | Writing | | | Emotion | | | ChestX | | |
|---|---|---|---|---|---|---|---|---|---|---|---|---|---|---|---|---|---|---|
| **Method** | 3-shot | 5-shot | 10-shot | 3-shot | 5-shot | 10-shot | 3-shot | 5-shot | 10-shot | 3-shot | 5-shot | 10-shot | 3-shot | 5-shot | 10-shot | 3-shot | 5-shot | 10-shot |
| ProtoNet (Chen et al., 2019) | 0.51 | 0.46 | 0.50 | 0.44 | 0.49 | 0.52 | 0.51 | 0.56 | 0.59 | 0.69 | 0.79 | 0.73 | 0.51 | 0.49 | 0.55 | 0.22 | 0.23 | 0.25 |
| DAPN (Zhao et al., 2021) | 0.49 | 0.51 | 0.55 | 0.57 | 0.61 | 0.63 | 0.46 | 0.49 | 0.51 | 0.75 | 0.79 | 0.80 | 0.48 | 0.50 | 0.59 | 0.24 | 0.24 | 0.26 |
| MAML (Chen et al., 2019) | 0.37 | 0.42 | 0.42 | 0.40 | 0.58 | 0.66 | 0.37 | 0.41 | 0.39 | 0.32 | 0.40 | 0.35 | 0.22 | 0.27 | 0.31 | 0.22 | 0.20 | 0.22 |
| SemiCMT (Chen et al., 2024b) | 0.33 | 0.39 | 0.49 | 0.37 | 0.40 | 0.44 | 0.24 | 0.34 | 0.43 | 0.27 | 0.39 | 0.48 | 0.24 | 0.34 | 0.47 | 0.25 | 0.26 | 0.28 |
| GPT2 (Zhou et al., 2023) | 0.11 | 0.09 | 0.12 | 0.13 | 0.09 | 0.09 | 0.13 | 0.14 | 0.17 | 0.14 | 0.19 | 0.20 | - | - | - | - | - | - |
| SRR-w/o-Removal | 0.42 | 0.47 | 0.52 | 0.44 | 0.46 | 0.47 | 0.35 | 0.37 | 0.48 | 0.48 | 0.57 | 0.64 | 0.26 | 0.31 | 0.38 | 0.16 | 0.18 | 0.19 |
| SRR | 0.45 | 0.55 | 0.64 | 0.68 | 0.66 | 0.71 | 0.45 | 0.47 | 0.62 | 0.55 | 0.76 | 0.73 | 0.31 | 0.34 | 0.53 | 0.22 | 0.21 | 0.21 |
| Our-Single | **1.42** | **1.57** | **2.09** | **2.76** | **2.61** | **2.51** | **1.55** | **1.90** | **1.76** | **2.11** | **1.88** | **1.84** | **0.82** | **1.07** | **1.26** | **0.83** | **1.04** | **1.24** |
| MDDA (Zhao et al., 2020) | 0.03 | 0.04 | 0.04 | 0.08 | 0.08 | 0.08 | 0.02 | 0.03 | 0.03 | 0.03 | 0.02 | 0.02 | 0.03 | 0.03 | 0.03 | 0.04 | 0.04 | 0.04 |
| MCW (Lee et al., 2019) | 0.06 | 0.05 | 0.05 | 0.06 | 0.07 | 0.06 | 0.04 | 0.05 | 0.05 | 0.06 | 0.06 | 0.06 | 0.05 | 0.05 | 0.05 | 0.04 | 0.04 | 0.05 |
| Our-Multi | **1.71** | **1.47** | **1.64** | **3.00** | **2.70** | **2.75** | **1.20** | **1.25** | **1.63** | **1.93** | **1.96** | **1.53** | **0.96** | **1.23** | **1.73** | **0.68** | **1.24** | **0.83** |

*Table 5.* Comparison in ATR ratio. Top-2 results are highlighted in **bold**

by 1.6 to 29 times and 16.6 to 98 times, respectively.

Without LWS, both *SRR-w/o-Removal* and *SRR* still outperform the baselines in accuracy on average for HHAR, WESAD, and Gesture, confirming that SRR alone makes a meaningful contribution under the cross-modality FSL settings. However, their gains are not consistent across all target datasets and they still remain below the oracle overall, which highlights the importance of LWS. In addition, *SRR* outperforms *SRR-w/o-Removal* on most target datasets, indicating that the layer channel removal further improves repairing performance. However, *SRR* yields slightly lower average accuracy on HHAR and Writing, suggesting that the effect of layer channel removal may be less stable when LWS is not enabled. This indicates that the switch control (*i.e.*, dynamically turning the removal on or off) could be further optimized across different target datasets. Moreover, *SRR-w/o-Removal* yields consistently lower ATR than the baselines. This suggests that, although the layer-wise connectors are lightweight, they still introduce additional resource costs and may hinder edge efficiency when no further structural optimization is applied.

After enabling LWS, both accuracy and ATR of *Our-Single* improve significantly, showing that LWS makes a major contribution by further improving both performance and efficiency on top of SRR. Although our variants remain slightly below the oracle on HHAR, Gesture, and Emotion under the 3-shot setting, our source-impact study shows that XTransfer can be further improved by reusing source models with higher quality or closer semantic relevance. Overall, XTransfer achieves SOTA accuracy while substantially reducing data and resource costs, providing a scalable and adaptable method for human sensing at the edge. Further results on training and on-device performance are provided in Appendix C.

## 7. Conclusion

This paper presents XTransfer, a novel method that enables modality-agnostic few-shot model transfer with resource-efficient design. XTransfer reuses the power of pre-trained models as free sources and requires only few sensor data to restructure models. It significantly reduces the costs associated with sensor data collection, model training, and edge deployment, offering a scalable and adaptable method for advancing human sensing applications.

**Discussion and future work.** XTransfer is motivated by the growing availability of public pre-trained models across modalities, and our experiments cover diverse sources from image, audio, text, and sensing modalities. Since it does not rely on shared semantic spaces or paired cross-modal data, it can support diverse sensing modalities whenever suitable source models are available. While XTransfer is designed to maximize the utility of the given source models, identifying the optimal source models for a target task remains another important direction for future work.

# Acknowledgements

This work was supported in part by the Faculty of Science and Engineering (FSE) Start-up Grant and the Macquarie University JUMPSTART Grant Program.

# Impact Statement

This paper proposes a pioneering and scalable method that enables modality-agnostic few-shot model transfer for advancing human sensing on edge systems. By leveraging publicly available pre-trained models and only few sensor data, it significantly reduces the costs of data collection and resources required to adapt to new sensing modalities. In practice, it enables fast prototyping and broad access to human sensing applications by lowering the barrier to adapting models without collecting large-scale sensor datasets. Our sensing data collection involving human participants was approved by our institution's Human Research Ethics Committee (Ref. No. 520221141241381), and all participants provided written informed consent. Collected data were anonymized at recording time with no personally identifiable information stored, and are securely retained and shared only under controlled conditions to protect participant privacy.

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

# Appendix

The appendix is organized as follows:

- Section A illustrates more details on our experiment setup, including implementation details, developed real-world sensing applications, experimental setups, dataset statistics, evaluation metrics, and computation resources we used for our experiments.
- Section B introduces additional technique details of different components in XTransfer.
- Section C gives more experimental results about ablation study and benchmarking.

# A. Experiment setup details

## A.1. Settings for learning with few data

A typical FSL method consists of two stages–meta-training and meta-testing (Chen et al., 2019). In meta-training stage, FSL requires a large-scale source dataset from the same modality (*i.e.*, typically specific to the application) to train source models from scratch. The source dataset is divided into a support set and a query set [2]. Data for each set is loaded according to the *n-way* (*i.e.*, number of classes) and *n-shot* (*i.e.*, number of samples per class) format (Song et al., 2023). The meta-testing stage follows the same structure but uses unseen target data. Specifically, the target support set is used to *adapt* the source models, which are then evaluated on the target query set. Leave-One-Out Cross-Validation (LOOCV) is widely used as the *de facto* standard for evaluating datasets involving multiple users in human sensing (Gong et al., 2019). LOOCV works by leaving one user out as the target for meta-testing while using the remaining users to train the source models during meta-training in each round. The MetaSense (Gong et al., 2019) is used as an oracle baseline under FSL settings, where sufficient target sensing datasets are provided during meta-training stage. Unlike FSL, cross-domain FSL focuses primarily on meta-testing and transfers pre-trained source models across domains (*i.e.*, different users), while remaining within the same modality (Oh et al., 2022). In this work, the cross-modality FSL settings are defined similarly to cross-domain FSL, but extend it to transferring pre-trained source models across different modalities, using only few labeled target data and without assuming any additional unlabeled target data.

## A.2. Setups in preliminary study

As shown in Figure 1(a), we test a standard *overfitting* metric, defined as $\frac{|Accuracy_{train} - Accuracy_{test}|}{Accuracy_{train}}$ (López et al., 2022). In Figure 1(b), we apply a standard cross-domain FSL benchmarking (Oh et al., 2022) to gain insights. It uses Earth Mover's Distance (EMD) to measure similarity between source models and targets. We test 3 source models (using a pre-trained ResNet18 from PyTorch Hub (PyTorch, 2026)) from 3 modalities, *e.g.*, miniImageNet (Image), Newsgroup (Text), and OPPORTUNITY (Sensing), shown in Table 2. We also use 5 sensing datasets (HHAR, WESAD, Writing, Emotion, and Gesture) as targets, and employ 2 well-known image datasets (CUB and ChestX) as reference targets to the Image, detailed in Table 2. It also assesses FSL difficulty by fine-tuning source models on targets under 5-shot settings with LOOCV enabled. Since source models usually involve a large number of classes (*e.g.*, 100 classes in miniImageNet), we propose a class pairing strategy that pairs source and target classes exhibiting similar latent feature distributions across modalities.

## A.3. Setups in layer-wise analysis

In Section 3, we conduct a layer-wise analysis to gain insights into cross-modality model transfer with few sensor data. To set up, we continue to use the standard ResNet18 (PyTorch, 2026) and segment it as 9 L-Blocks, as shown in Table 3. We also employ the default reshaping (see Section B.1) on HHAR as target with 5-shot setting, and use the original source model input (miniImageNet) as reference, marked as Original. We examine the single-source baselines, including Transfer Learning (TL) (Jiang et al., 2022), Fine-tuning (FT) (Oh et al., 2022), MAML (Chen et al., 2019), DAPN (Zhao et al., 2021), and MetaSense (Gong et al., 2019) as the oracle baseline, shown in Table 1. To precisely track the MMC shift changes at each layer during fine-tuning,

---

[2]The support set and query set are assumed to be drawn from the same data distribution (Triantafillou et al., 2020).

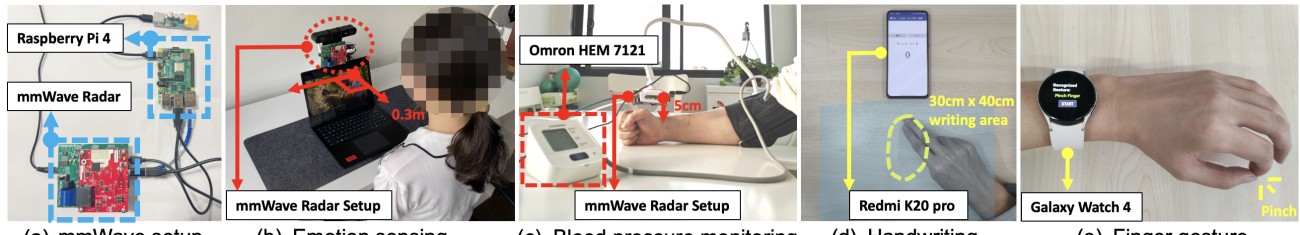

(a) mmWave setup    (b) Emotion sensing    (c) Blood pressure monitoring    (d) Handwriting    (e) Finger gesture

*Figure 5.* (a) Embedded mmWave radar testbed setup; (b)-(e) Built human sensing applications across different real-world settings.

### A.4. Layer-wise metric

To precisely track the MMC shift changes at each layer during fine-tuning, we essentially use the Silhouette score (S-score) (Shahapure & Nicholas, 2020) as the layer-wise metric to evaluate how well the features are being fine-tuned. The S-score is in the range from -1 to 1 and being better if closing to 1. To verify whether the S-score indicates the layer performance, we observe the changes in the S-score and accuracy at each layer using the TL (Jiang et al., 2022) and Original methods. Figure 6(a) also shows that S-score and accuracy by Original are positively correlated, converging in a layer-wise manner. TL's S-score also performs equivalently to its accuracy (*i.e.*, S-score drop along with accuracy drop) in detecting the negative transfer across different layers.

### A.5. Implementation

We implement `XTransfer` using PyTorch version 1.4.0. We also benchmark `XTransfer` with a full development cycle of cross-modality FSL tasks, including pre- and post-processing, training, and inference on commercial edge devices. Once pre-trained source models and few sensor data are processed in the cloud via `XTransfer`, compact models can be seamlessly converted to various DL platforms (*e.g.*, PyTorch Mobile, TFLite) for broader edge deployments. To enable reproducibility of the results, we fix all random seeds in the code, set the evaluation mode consistently when evaluating output models, and enable CUDA convolution determinism. To measure model resource overhead (*e.g.*, FLOPs and parameters), we use the standard PyTorch OpCounter. For regression tasks, class-based transformation (Shi et al., 2022) enables compatibility.

### A.6. Developed sensing applications

To evaluate `XTransfer` on real-world human sensing targets, we essentially develop 4 human sensing applications, as shown in Figure 5. These include: (1) emotion recognition via facial expressions (*i.e.*, Emotion) and (2) non-contact blood pressure monitoring (*i.e.*, BP) based on an embedded mmWave radar setup, (3) handwriting tracking (*i.e.*, Writing) using ultrasound based on a smartphone, and (4) gesture recognition (*i.e.*, Gesture) using IMU and PPG sensors based on a smartwatch. The data collection is approved with ethics approval and involves 40 subjects. The corresponding target sensing datasets are shown in Table 2 with detailed specifications.

### A.7. Hardware setup and computation environment

We evaluate `XTransfer` on real-world testbeds using 3 commercial edge devices, ranging from smartwatch to Raspberry Pi (Table 6). For example, Figure 5(a) presents an embedded mmWave radar setup which consists of a TI IWR1843Boost and a Raspberry Pi 4 using ROS Melodic on Ubuntu 18.04.4. In addition, we implement both smartwatch and smartphone testbeds using PyTorch Android version 1.10.0. We also employ servers equipped with NVIDIA GeForce RTX 3090 GPU to build our cloud environment.

| Device | ROM | RAM | CPU | Battery | OS |
|---|---|---|---|---|---|
| Galaxy Watch 4 | 16GB | 1.5GB | Exynos W920 | 247mAh | Wear OS 3.5 |
| Raspberry Pi 4 | 64GB | 8GB | Broadcom BCM2711 | - | Ubuntu 18.04.4 |
| Redmi K20 Pro | 128GB | 8GB | Snapdragon 855 | 4000mAh | Android 11 |

*Table 6.* Edge devices specification.

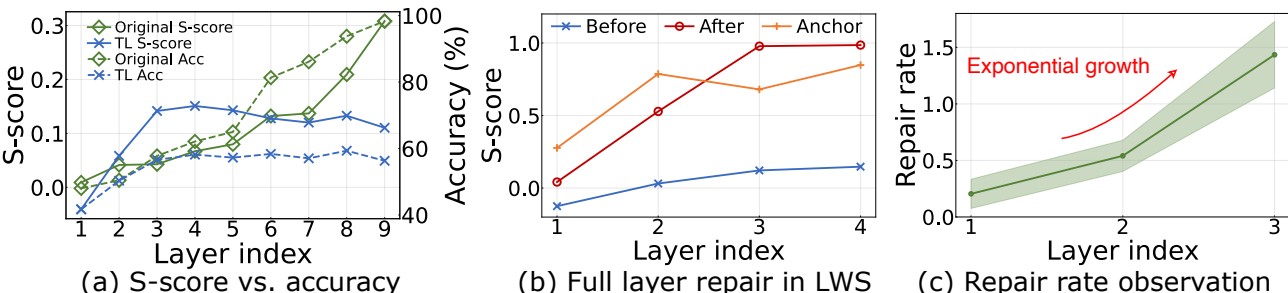

*Figure 6.* Design insights. **(a)** shows layer-wise metric correlation. **(b)(c)** present efficient search insights into LWS control using multiple source models.

### A.8. Training details

In the Repair stage, we fine-tune the generative transfer module (see Section 4) for 100 episodes using SGD with a learning rate of 1e-2 and momentum 0.95. We apply a step-based learning rate scheduler with a step size of 20 and a decay factor of 0.5. We also use standard early stopping with a patience of 20. We retain the training parameters of the baselines in Table 1 as specified in their original implementations.

## B. Technical details

### B.1. Default reshaping

To align with source model input shape, we develop a *default reshaping* to transform sensor data shape. It uses bilinear interpolation (Mastyło, 2013) (*i.e.*, Resizer) to resize the height and width, and a fixed convolutional layer (*i.e.*, Fixed header) with a $1 \times 1$ kernel to match the input channels (Szegedy et al., 2014), placed between source models and sensor data.

### B.2. Setups for efficient search insights

To gain design insights of efficient search for LWS control (see Section 5), we observe a complete process where all layers are repaired using the HHAR dataset and a Multi-ResNet18 model (*i.e.*, 5 source models with 45 L-unit), as shown in Table 3. Figure 6(b) shows the changes in the S-score across three states: after pairing anchors (*i.e.*, Anchor), before and after applying the SRR pipeline (*i.e.*, Before and After, respectively), in a layer-wise manner. The results show that the SRR pipeline successfully increases the S-score from Before state (*i.e.*, lower bound) to align with the Anchor state (*i.e.*, the objective). This indicates a strong correlation between the After state's S-score and both Before and Anchor values. These insights lead to the observation for designing the repair rate growth model.

### B.3. Observation for designing repair rate growth model

Our key idea is to estimate the repaired S-score (*i.e.*, repair value) of each layer *before* initiating SRR processing, allowing us to decide whether unnecessary repairs can be bypassed. To empirically examine this correlation, we define a *repair rate function* at the $n$-th recombined layer as the repair value gain normalized by the paired anchor S-score:

$$rate_n = \frac{\text{After} - \text{Before}}{\text{Anchor}} = \frac{V_{ij}^n - \text{Score}(\text{Pro}(ft_{ij}^n), Y_t)}{\text{Score}(\text{Pro}(fs_{ij}^n), Y_s)},$$

where $\text{After} = V_{ij}^n$ denotes the repair value, $\text{Before} = \text{Score}(\text{Pro}(ft_{ij}^n), Y_t)$ denotes the S-score before repair, $\text{Anchor} = \text{Score}(\text{Pro}(fs_{ij}^n), Y_s)$ denotes the paired anchor score, with $Y_s$ being the source labels. We normalize $rate_n$ to $[0, 1]$ and repeat this computation multiple times to examine its trend. Figure 6(c) shows that the repair rate exhibits an *exponential growth* pattern. Based on this observation, we design the repair rate growth model using an exponential function (see Section 5.2).

### B.4. Post fine-tuning

After LWS control is completed, we add either a linear classifier or a regression layer (*e.g.*, a fully connected layer) at the end of the model to finalize its backbone. To strengthen layer-wise dependence, we perform a quick post fine-tuning process, updating only the connectors and fully connected layers.

| Epoch (#) | #1 | #20 | #40 | #60 | #80 | #100 |
|---|---|---|---|---|---|---|
| With Alignment | 0.486 | 0.355 | 0.350 | 0.339 | 0.333 | 0.331 |
| Without Alignment | 0.476 | 0.406 | 0.393 | 0.383 | 0.378 | 0.387 |

*Table 7.* Feature space alignment performance across training epochs.

| PCA Variation | 3-shot | | 5-shot | | 10-shot | |
|---|---|---|---|---|---|---|
| | Accuracy (%) | Time (mins) | Accuracy (%) | Time (mins) | Accuracy (%) | Time (mins) |
| Standard PCA-#2 | 58.96 | 3.9 | 71.85 | 5.6 | 80.74 | 10.3 |
| Sparse PCA-#2 | 58.52 | 3.8 | 64.67 | 5.3 | 75.11 | 8.8 |

*Table 8.* Impact of PCA variations on accuracy and training time.

# C. Additional experiment results

### C.1. Feature space alignment performance

In the process of feature space alignment (Section 4.1), we use a standard Hungarian algorithm based on default distance metrics to select and pair classes. To verify its effectiveness, we keep using the setup of HHAR, the standard pre-trained ResNet18 on miniImageNet, and the standard PCA under 5-shot settings to examine how our repair loss performs with/without the feature space alignment, especially in the initial layer repairing. Repairing the initial layer is critical, as it often introduces a large, disorganized feature space and significantly affects the repair of subsequent layers. In Table 7, the results show that with the alignment, the repair loss decreases considerably faster by 1.87 times on average, compared to the case without alignment. This indicates that the design of feature space alignment notably contributes to the repairing process.

### C.2. Impact of PCA variations

Since our method uses the repair learning mechanism that relies on backpropagation (BP) through the reduced-orthogonal feature space, standard PCA can project data into a linear subspace and maintain a closed-form, differentiable structure, which enables smooth integration with BP in our end-to-end learning pipeline. Compared to advanced choices such as kernel PCA, we find that it introduces a non-linear mapping into a high-dimensional feature space using a kernel function, and its projections are generally not differentiable with respect to input features. This makes it incompatible with our pipeline. To test whether linear PCA variations affect the overall performance of XTransfer, we compare standard PCA against sparse PCA using the setup of HHAR based on the ResNet18 under 3 few-shot settings with a default search depth of 3. As shown in Table 8 (#2 denotes 2-component setting), sparse PCA reduces learning time slightly by 11.6% on average, but suffers from a notable accuracy drop by 4.4% on average. These results support the choice of standard PCA as a balanced and practical solution.

### C.3. Impact of PCA components

We then evaluate the impact of PCA components on the model accuracy and time efficiency of XTransfer. Since the feature space with different dimensionalities adjusted by PCA components may impact our method performance, we test 3 different levels of PCA components, including PCA-#2 (top two components), PCA-80% (selected components covering 80% of the feature variance), and PCA-Max (maximum components available). We continue to use the same setup as above with the standard PCA. The results in Table 9 show that as the number of PCA components increases, the training time rises notably. In fact, PCA-Max results in an average accuracy drop of 4.5% compared to PCA-80%. By contrast, PCA-#2 achieves the best accuracy-to-time ratio across all n-shot settings, with competitive accuracy and the lowest training time, hence we use PCA-#2 as the default throughout the evaluation.

| PCA Components | 3-shot | | 5-shot | | 10-shot | |
|---|---|---|---|---|---|---|
| | Accuracy (%) | Time (mins) | Accuracy (%) | Time (mins) | Accuracy (%) | Time (mins) |
| PCA-#2 | 58.96 | 3.9 | 71.85 | 5.6 | 80.74 | 10.3 |
| PCA-80% | 63.61 | 5.2 | 74.72 | 7.6 | 80.00 | 12.9 |
| PCA-Max | 60.00 | 7.2 | 75.56 | 8.1 | 70.28 | 16.9 |

*Table 9.* Impact of PCA components on accuracy and training time.

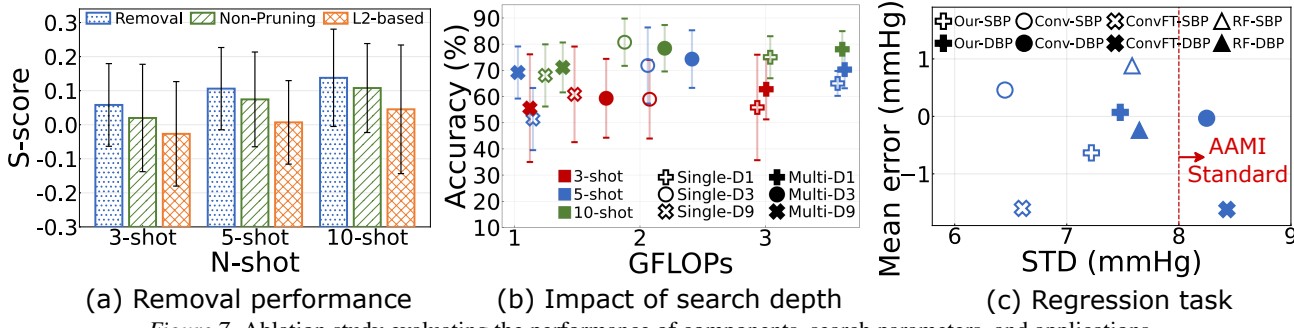

*Figure 7.* Ablation study evaluating the performance of components, search parameters, and applications.

| Alignment Method | 3-shot | | 5-shot | | 10-shot | |
|---|---|---|---|---|---|---|
| | Accuracy (%) | ATR | Accuracy (%) | ATR | Accuracy (%) | ATR |
| MMD Loss (Wang et al., 2023) | 55.78 | 1.63 | 65.89 | 2.01 | 68.52 | 1.83 |
| Repair Loss | 65.11 | 2.33 | 69.33 | 2.45 | 72.07 | 2.29 |
| PCA + Repair Loss | 58.96 | 1.42 | 71.85 | 1.56 | 80.74 | 2.09 |

*Table 10.* Comparison of non-linear feature alignment methods across n-shot settings.

## C.4. Non-linear feature alignment performance

We essentially use PCA to construct a fixed, orthogonal anchor feature space in which alignment is measured and the anchor-based repair loss is computed at each layer across modalities. The proposed connector (*i.e.*, repair generator) dynamically maps target-modality features into this anchor space, enabling the non-linear feature alignment. To examine the impact of using a linear PCA anchor space and to compare the alignment performance, we evaluate our repair loss with and without PCA against a Maximum Mean Discrepancy (MMD) loss baseline (Wang et al., 2023). To set up, we keep using the standard ResNet18 on miniImageNet (Table 3) as sources on HHAR as target (Table 2) in 3-, 5- and 10-shot settings with all configurations identical except for the alignment loss. As shown in Table 10, our repair loss with PCA anchor space achieves the highest accuracy of 70.5% and the lowest ATR of 1.69 on average across all n-shot settings. Compared to our repair loss without PCA, MMD loss performs a lower accuracy on average. This indicates that MMD loss is limited to aligning the latent feature distribution when the feature space is highly complex due to strong cross-modality discrepancies and only few target data. In addition, adversarial alignment methods such as DAPN (Zhao et al., 2021) and MDDA (Zhao et al., 2020) also face similar difficulties under these conditions, as shown in Tables 4 and 5. Moreover, the PCA anchor space provides a more stable basis for alignment and consistently enhances our repair loss performance. As a result, these findings further validate the effectiveness of our method in cross-modality FSL settings.

## C.5. Layer-wise convergence comparison

To further examine layer-wise convergence across the above alignment methods, we observe the feature alignment process in the initial layer. To normalize heterogeneous losses, we follow the strategy in (Li et al., 2020) and compute the loss descent (LD) relative to the initial loss, enabling a fair comparison of convergence rates across different loss functions (*i.e.*, how much each loss decreases from its starting point), defined as $\mathrm{LD}(t) = (\mathcal{L}(0) - \mathcal{L}(t))/\mathcal{L}(0) \times 100\%$, where $t$ denotes the epoch index. All experimental settings remain the same as in the previous setup. As shown in Table 11, our repair loss with the PCA anchor achieves the largest loss descent of 28.87%, which is 1.79 and 1.87 times faster than the repair loss without PCA and the MMD loss, respectively. Interestingly, MMD loss presents a large initial drop but quickly plateaus, while our repair loss starts more gradually and accelerates over training. In contrast, the PCA-based repair maintains a steady and consistent descent across epochs, highlighting the benefits of aligning features in a fixed, orthogonal anchor space. These findings further support that our method provides both effective feature alignment and practical training efficiency, consistent with the efficiency study in Section C.11.

## C.6. Removal performance

We next evaluate how layer channel removal enhances the average S-score of selected layers. We continue to use the same setup as above under 5-shot settings. We employ L2 norm-based (*i.e.*, L2-based) model pruning (Frankle & Carbin, 2019; Shen et al., 2022) as the baseline and compare it with the setting where layer channel removal is disabled (*i.e.*, Non-Pruning).

| Epoch (#) | #10 | #20 | #30 | #40 | #50 | #60 | #70 | #80 | #90 | #100 |
|---|---|---|---|---|---|---|---|---|---|---|
| MMD Loss (Wang et al., 2023) | 6.46% | 10.0% | 11.63% | 12.81% | 13.5% | 14.3% | 14.66% | 14.98% | 15.27% | 15.41% |
| Repair Loss | 0.03% | 1.67% | 2.66% | 4.93% | 5.26% | 7.16% | 8.06% | 10.4% | 14.66% | 16.14% |
| PCA + Repair Loss | 5.12% | 13.45% | 16.54% | 19.23% | 20.55% | 21.3% | 25.96% | 26.21% | 27.6% | 28.87% |

*Table 11.* Comparison of layer-wise convergence rates across alignment methods.

| Backbone | Accuracy (%) | FLOPs (G) |
|---|---|---|
| Multi-Conv4 | 65.56 | 0.01 |
| Multi-ResNet10 | 63.61 | 0.14 |
| Multi-ResNet18 | 78.44 | 2.19 |

*Table 12.* Performance of different source model backbones in terms of accuracy and FLOPs.

Figure 7(a) demonstrates that the proposed layer channel removal outperforms both Non-Pruning and L2-based pruning across all settings, achieving an average S-score of 0.1 by selected layers. L2-based pruning achieves a lower average S-score than Non-Pruning, indicating that the layer channel importance may be inaccurately estimated, resulting in degraded performance. These results show that the proposed layer channel removal further improves layer performance by accurately removing unnecessary layer channels at each layer.

### C.7. Impact of search depth

In this experiment, we investigate how different search depths affect the accuracy and resource overhead (*e.g.*, FLOPs) of output models. Varying the search depth alters the search window sizes with the number of layer candidates, which can lead to different output model backbones and accuracy. To quantify these differences, we employ HHAR as target, ResNet18 on miniImageNet (Single) and Multi-ResNet18 (Multi) on 5 image datasets as sources (shown in Table 2) in 3-, 5- and 10-shot settings. We also utilize MetaSense in Table 1 as the oracle baseline. Both Single and Multi models have up to 9 L-unit, hence we set three different search depths as D1, D3, D9. Figure 7(b) shows that the output models of both Single-D1 and Multi-D1 result in high FLOPs, while that of both Single-D9 and Multi-D9 achieve low FLOPs. As search depth decreases, the search process becomes longer, and it triggers LWS control to make more decisions on recombining layers, which results in a suboptimal model backbone with high FLOPs. Although both Single-D3 and Multi-D3 do not produce the lowest FLOPs on average, they outperform others and achieve the highest average model accuracy of 70.52% and 70.69%, respectively, across the three settings. Therefore, we set D3 as the default search depth, as it provides the best balance between resource overhead and accuracy of the output models.

### C.8. Impact of model backbones

The current implementation of XTransfer uses homogeneous pre-trained source models. Technically, the layer-wise repairing and recombination techniques impose no fundamental restrictions on the choice of model backbone and work with various architectural variants. To verify this, we evaluate the trade-off between accuracy and computational cost using different source backbones. We continue to use HHAR, and test Multi-Conv4, Multi-ResNet10, and Multi-ResNet18 in a 5-shot, multi-source setting shown in Table 3. The results in Table 12 demonstrate that the average accuracy declined, transitioning from Multi-Conv4 to Multi-ResNet10, and then increasing with Multi-ResNet18. Interestingly, Multi-Conv4, despite having fewer parameters, outperforms Multi-ResNet10, which may be attributed to differences in model quality or distributional divergence between source and target. Multi-ResNet18, though with higher FLOPs, achieves the best accuracy. It suggests that larger models trained on large-scale cross-modality datasets exhibit superior latent features, which notably contribute to such model transfer. Our techniques such as the connectors are designed to flexibly handle the layers with different shapes/structures, and can be extended to heterogeneous model backbones. In addition, our method focuses on latent feature alignment rather than specific feature extraction. It remains fully compatible with advanced source models (*e.g.*, with temporal encoders), which can be seamlessly integrated into our pipelines with minimal or no modifications.

### C.9. Regression task performance

We also evaluate the effectiveness of XTransfer in a regression task using the BP dataset, which includes both systolic (SBP) and diastolic (DBP) sensor data shown in Table 2 collected by the blood pressure monitoring application. We employ the pre-trained ResNet18 on miniImageNet as the source model and use the 5-shot setting. To integrate this into

| Method | 3-shot | | 5-shot | | 10-shot | |
|---|---|---|---|---|---|---|
| | Accuracy (%) | ATR | Accuracy (%) | ATR | Accuracy (%) | ATR |
| DAPN (Zhao et al., 2021) | 48.7 | 0.49 | 51.2 | 0.51 | 54.7 | 0.55 |
| DAPN+Pruning (Shen et al., 2022) | 25.0 | 0.66 | 28.3 | 0.64 | 30.0 | 0.77 |
| SRR-w/o-Removal | 58.0 | 0.42 | 63.6 | 0.47 | 71.2 | 0.52 |
| SRR | 53.7 | 0.45 | 61.8 | 0.55 | 71.6 | 0.64 |
| Our-Single | 59.0 | 1.42 | 71.8 | 1.57 | 80.7 | 2.09 |

*Table 13.* Comparison with post-hoc compression.

| | HHAR (mins) | | | WESAD (mins) | | | Gesture (mins) | | | Writing (mins) | | | Emotion (mins) | | | ChestX (mins) | | |
|---|---|---|---|---|---|---|---|---|---|---|---|---|---|---|---|---|---|---|
| Method | 3-shot | 5-shot | 10-shot | 3-shot | 5-shot | 10-shot | 3-shot | 5-shot | 10-shot | 3-shot | 5-shot | 10-shot | 3-shot | 5-shot | 10-shot | 3-shot | 5-shot | 10-shot |
| DAPN (Zhao et al., 2021) | 2.38 | 4.62 | 11.78 | 1.49 | 2.54 | 6.78 | 3.36 | 6.12 | 15.72 | 4.12 | 7.71 | 20.97 | 2.99 | 7.34 | 18.86 | 2.14 | 4.57 | 12.04 |
| MAML (Chen et al., 2019) | 124.28 | 125.66 | 183.20 | 71.71 | 77.07 | 115.87 | 118.79 | 129.79 | 191.95 | 165.06 | 129.27 | 194.36 | 94.81 | 98.40 | 153.20 | 66.20 | 72.17 | 106.97 |
| SemiCMT (Chen et al., 2024b) | 3.02 | 8.27 | 10.23 | 1.82 | 2.48 | 5.63 | 4.81 | 9.42 | 12.83 | 10.10 | 13.92 | 18.75 | 3.39 | 7.32 | 11.48 | 2.84 | 5.28 | 8.33 |
| Our-Single | 3.98 | 5.57 | 7.95 | 2.12 | 2.99 | 4.33 | 5.64 | 7.80 | 11.24 | 7.94 | 10.54 | 14.69 | 5.58 | 7.43 | 9.88 | 3.65 | 4.75 | 6.84 |
| Our-Multi | 9.25 | 15.33 | 15.25 | 4.40 | 5.38 | 7.21 | 12.49 | 15.12 | 24.37 | 15.35 | 17.36 | 32.38 | 11.72 | 13.39 | 19.75 | 8.41 | 9.33 | 13.61 |

*Table 14.* Comparison in training time of each method.

the SRR pipeline, we transform the regression task by treating individual regression labels as distinct classes for anchor class pairing. For benchmarking, we use mean error (ME) and standard deviation (STD), following the Advancement of Medical Instrumentation (AAMI) standard, which accepts an ME within 5 mmHg and STD within 8 mmHg (Stergiou et al., 2018). We employ a random forest (RF) based method (Shi et al., 2023) as the oracle baseline. Besides, we train a Conv4 backbone in Table 3 (Conv) using LOOCV, and the fine-tuned Conv (ConvFT) as the baselines. Figure 7(c) shows that both XTransfer and the RF methods surpass the AAMI standard. In particular, XTransfer outperforms the baselines and RF in estimating both SBP (ME: -0.64mmHg, STD: 7.2mmHg) and DBP (ME: 0.07mmHg, STD: 7.47mmHg) tasks. These results demonstrate that XTransfer is highly efficient in handling regression tasks with few sensor data.

### C.10. Post-hoc compression comparison

To examine whether a strong baseline combined with post-hoc compression may maintain accuracy and resource efficiency under our setting, we conduct a post-hoc compression comparison on DAPN (Zhao et al., 2021). Since LWS is not directly compatible with DAPN, we instead apply the L2-based structural pruning (Shen et al., 2022) to DAPN. For a fair comparison, we set the pruning rate such that the pruned DAPN matches the average model size of Our-Single under each shot setting. We then report both Accuracy and ATR on HHAR using the same experimental setup as Our-Single in Section 6.3 across the 3-, 5-, and 10-shot settings. As shown in Table 13, although pruning improves ATR, it causes a substantial loss in accuracy. Averaged over the three shot settings, DAPN drops from 51.5% to 27.8% in accuracy, while ATR increases from 0.52 to 0.69. This result suggests that simply applying post-hoc compression to the baseline can easily fail to maintain performance under the cross-modality FSL settings, which further motivates the need for XTransfer with LWS-enabled restructuring.

### C.11. Training efficiency

To test the practical training efficiency in the cloud, we further evaluate training time across the selected baselines. We use the same experimental setups as in Table 15, Section 6.3. For MAML (Chen et al., 2019) and SemiCMT (Chen et al., 2024b), which require model re-training from pre-trained source models to adapt to the target modality classes, training time includes both re-training and fine-tuning. For DAPN (Zhao et al., 2021), it reflects model adaptation process on few target data. The results in Table 14 show that both Our-Single and Our-Multi achieve significant training time reductions compared to MAML. Our-Single also matches or exceeds DAPN and SemiCMT in training efficiency. Notably, even when using up to 5 source models, Our-Multi remains practical training efficiency, requiring only 2.05 times more training time on average compared to Our-Single across all datasets and shot settings. This further supports the practicability and scalability of our method.

### C.12. Training resource usage

We then examine the training resource usage of XTransfer by measuring GPU utilization (Util) in the cloud. We focus on the above baselines that require model re-training or adaptation for transfer, *e.g.*, MAML (Chen et al., 2019), DAPN (Zhao et al., 2021), and SemiCMT (Chen et al., 2024b)), excluding methods that rely solely on fine-tuning or distance-based techniques. The experimental setup remains the same as above. As shown in Table 15, both our methods achieve the lowest

| Method | HHAR (%) | | | WESAD (%) | | | Gesture (%) | | | Writing (%) | | | Emotion (%) | | | ChestX (%) | | |
|---|---|---|---|---|---|---|---|---|---|---|---|---|---|---|---|---|---|---|
| | 3-shot | 5-shot | 10-shot | 3-shot | 5-shot | 10-shot | 3-shot | 5-shot | 10-shot | 3-shot | 5-shot | 10-shot | 3-shot | 5-shot | 10-shot | 3-shot | 5-shot | 10-shot |
| DAPN (Zhao et al., 2021) | 48.85 | 49.19 | 51.79 | 44.62 | 49.49 | 48.81 | 42.73 | 48.71 | 50.59 | 43.88 | 56.63 | 57.51 | 44.67 | 59.73 | 63.12 | 46.66 | 47.32 | 48.48 |
| MAML (Chen et al., 2019) | 59.59 | 79.32 | 89.72 | 52.58 | 70.27 | 77.52 | 53.11 | 66.75 | 73.21 | 72.56 | 85.47 | 92.22 | 63.39 | 72.78 | 75.01 | 44.52 | 55.46 | 61.36 |
| SemiCMT (Chen et al., 2024b) | 55.60 | 73.43 | 78.91 | 38.83 | 50.33 | 67.89 | 57.02 | 72.16 | 98.04 | 75.14 | 79.12 | 81.79 | 60.30 | 63.43 | 68.60 | 41.54 | 50.05 | 63.43 |
| Our-Single | **18.27** | **18.90** | **19.48** | **15.95** | **16.46** | **16.98** | **20.67** | **21.71** | **22.51** | **19.92** | **20.67** | **21.08** | **18.24** | **19.20** | **19.96** | **16.97** | **17.58** | **18.80** |
| Our-Multi | **18.31** | **20.77** | **22.06** | **18.00** | **19.42** | **23.10** | **20.49** | **22.49** | **24.49** | **18.68** | **19.25** | **21.69** | **16.45** | **18.18** | **19.85** | **15.33** | **17.69** | **19.62** |

*Table 15.* Comparison in GPU utilization of each method during training. Top-2 results are highlighted in **bold**

| Dataset | Type | FLOPs | Params | Watch | | Pi | | Phone | |
|---|---|---|---|---|---|---|---|---|---|
| | | | | Latency (ms) | Mem (MB) | Latency (ms) | Mem (MB) | Latency (ms) | Mem (MB) |
| HHAR | Single | 1.51G | 0.75M | 625.5 | 125.5 | 274.18 | 246.62 | 26.86 | 179.5 |
| | Multi | 1.46G | 2.16M | 634.5 | 140 | 249.26 | 247.83 | 28.34 | 209 |
| WESAD | Single | 1.19G | 0.74M | 604.5 | 120 | 240.05 | 244.64 | 25.77 | 168 |
| | Multi | 0.89G | 0.34M | 460.5 | 122 | 209.52 | 235.55 | 21.57 | 181 |
| Gesture | Single | 1.21G | 2.25M | 706.5 | 135 | 259.25 | 251.26 | 31.5 | 192 |
| | Multi | 1.45G | 2.09M | 692 | 137 | 268.79 | 253.7 | 30.96 | 190 |
| Writing | Single | 1.32G | 0.87M | 467.5 | 126 | 261.71 | 242.33 | 29.79 | 183.5 |
| | Multi | 1.28G | 0.73M | 513 | 124.5 | 285.2 | 247.01 | 34.67 | 185 |
| Emotion | Single | 1.51G | 0.49M | 767.5 | 130.5 | 390.36 | 271.02 | 41.1 | 171.5 |
| | Multi | 1.43G | 2.14M | 781 | 142 | 285.16 | 251.48 | 37.9 | 189.5 |
| ChestX | Single | 0.64G | 0.72M | 690.5 | 126 | 210.45 | 237.45 | 29.57 | 165 |
| | Multi | 1.06G | 0.7M | 670 | 130.5 | 261.14 | 236.96 | 30.14 | 183.5 |
| ResNet18 | - | 3.67G | 11.18M | 1060 | 189 | 703.52 | 303.75 | 89.8 | 311 |
| Multi-ResNet18 | - | 14.68G | 55.9M | 5816 | 356 | 4242.27 | 482.26 | 454.44 | 436 |

*Table 16.* On-device Performance.

GPU Util across all n-shot settings compared to baselines. Notably, both Our-Single and Our-Multi achieve a substantial reduction in GPU Util by 2.6 to 3.6 times and 2.5 to 3.5 times on average, respectively, compared to baselines. This is due to our layer-wise method, which progressively repairs and recombines only a subset of layers rather than training the entire model at once. It also highlights the effectiveness of our compact connector design. Despite leveraging multiple source models, Our-Multi shows no significant increase (*e.g.*, only 3.67%) in training resource usage, further highlighting its efficiency and scalability. Hence, XTransfer significantly reduces cloud resource usage, enabling scalable model transfer.

### C.13. On-device performance

We now evaluate the on-device performance of deploying XTransfer's output models on 3 commercial edge devices (Table 6), focusing on resource overhead metrics such as FLOPs, parameters, latency per call, and peak memory footprint. For each target dataset, we test both types of output models created by our single-source (Single) and multi-source (Multi) methods. We also use ResNet18 and Multi-ResNet18 (Table 3) as baselines. As shown in Table 16, all output models achieve significant reductions in model resource overhead, *e.g.*, FLOPs is reduced by 2.4 to 5.7× for Single models and by 10.1 to 16.5× for Multi models, compared to the baselines. These recombined models also demonstrate substantial improvement in on-device performance. Specifically, in the Single method, latency is reduced by 1.4 to 4.1 × and peak memory footprint decreases by 1.8 to 3.9×. In the Multi method, the reductions are even more pronounced, with latency dropping by 7.4 to 21× and memory footprint by 2.2 to 8.4×, compared to the baselines. These results prove that XTransfer successfully achieves streamlined edge deployment.

