# OpenReview forum: "XTransfer: Modality-Agnostic Few-Shot Model Transfer for Human Sensing at the Edge"
_ICML.cc/2026/Conference — ICML 2026 regular_

### Official Review · Reviewer_Bxtr · 2026-02-27

**Soundness:** 3
**Presentation:** 3
**Significance:** 3
**Originality:** 3
**Overall Recommendation:** 4
**Confidence:** 4

**Summary:**

This paper introduces XTransfer, a novel framework for transferring pretrained models from source modalities to target human sensing modalities using only few labeled sensor data. The core idea is to mitigate modality shift via layer-wise repair of latent feature distributions, using a Splice Repair Removal (SRR) pipeline and a Layer-Wise Search (LWS) control for efficient model restructuring. Experiments on 8 datasets and 4 real world sensing applications demonstrate consistent gains in accuracy and resource efficiency compared to SOTA baselines.

**Compliance With Llm Reviewing Policy:**

Affirmed.

**Key Questions For Authors:**

Please refer to the comments.

**Limitations:**

yes

**Strengths And Weaknesses:**

Strengths

1. The problem of cross-modality few-shot transfer is highly relevant for edge-based human sensing, where data is scarce and diverse.
2. The paper provides a thorough layer-wise analysis of modality shift, motivating the SRR pipeline and LWS control.
3. Experiments cover a wide range of source/target modalities, datasets, and real-world applications.

Weaknesses

1. The method assumes access to high-quality pretrained models from public repositories. While this is increasingly common, it may not hold for all sensing modalities or deployment scenarios.
2. The paper reports strong average performance but does not deeply analyze when and why the method fails—e.g., which source target pairs lead to negative transfer, or which layers are most difficult to repair.
3. One concern is about the modality-agnostic mechanism. In my view, different modalities have their own characteristics. If the model has no idea about the modality, how does it achieve high-quality modality transformation?
4. Writing and Presentation: Some sections are dense and could be better organized. Figures (e.g., Fig. 4) are cluttered and difficult to tell.

---

> ### Author Rebuttal · Authors · 2026-03-28
>
> We thank the reviewer for the positive assessment and for recognizing the importance of the problem, the value of the layer-wise analysis, and the breadth of the experiments. We address their questions below.
>
> **Q1: The method assumes access to high-quality pre-trained models from public repositories. How limiting is this assumption?**
>
> XTransfer is motivated by the growing availability of public pre-trained models across modalities, and our experiments cover diverse public pre-trained sources from image, audio, text, and sensing modalities. Since it does not rely on shared semantic spaces or paired cross-modal data, it is applicable to diverse sensing modalities whenever a suitable source model is available. Given such a source model, XTransfer is designed to make the best use of it and optimize transfer performance.
>
> Our source-impact study in Sec. 6.2 shows that different source models can lead to notably different outcomes. Sec. 6.3 further discusses this limitation and suggests that performance could be further improved with stronger or more semantically relevant source models. While XTransfer makes the best use of a given source model, finding the optimal source model remains a challenge beyond the current scope. We will make this dependence more explicit in the next revision.
>
> **Q2: When and why can XTransfer fail?**
>
> As suggested by Fig. 1(b), the performance of different source modalities is correlated with cross-modality similarity and few-shot difficulty, indicating that transfer becomes more fragile when similarity is low and the few-shot setting is harder. The results in Sec. 6.2 and 6.3 further support this trend.
>
> Negative transfer widely appears across different source layers and source–target pairs. This is why XTransfer combines SRR and LWS. SRR repairs misaligned layers, while LWS identifies which repaired layers are worth keeping. In the layer value function V_ij in Sec. 5.1, a low repaired value indicates that a layer is difficult to repair and is therefore unlikely to be selected by LWS. Our new ablation results with LWS disabled (see our response to Reviewer 1Bf2) further support this point, showing that retaining repaired low-value layers can cause a notable accuracy drop. We will make these limitations more explicit in the next revision.
>
>
> **Q3: If the model has no idea about the modality, how does it still achieve high-quality modality transformation?**
>
> Even if the source model has no direct knowledge of the target modality, its pre-trained discriminative latent representations can still be effectively reused if the target features can be transformed to align with the corresponding source latent representations, as supported by our layer-wise analysis in Sec. 3 and the anchor-based alignment design in Sec. 4.1. This is also consistent with recent work [1] showing that useful cross-modal structure can be leveraged even without explicit paired cross-modal data.
>
> This motivates our layer-wise repairing design, which optimizes latent feature alignment rather than relying on shared semantic spaces or paired cross-modal data. Each source layer provides a stable anchor distribution, and the connector is fine-tuned with the few target samples to transform sensing features toward the corresponding anchor features. Working together with LWS, which restructures the useful layers, XTransfer achieves high-quality transformation without relying on modality-specific semantic supervision.
>
> Overall, we will clarify the above limitations and failure modes more explicitly, and improve the clarity of the presentation, including figures such as Fig. 4.
>
> **References**
>
> [1] Gupta et al., Better Together: Leveraging Unpaired Multimodal Data for Stronger Unimodal Models. ICLR 2026.

---

> > ### Author Rebuttal · Reviewer_Bxtr · 2026-04-04
> >
> > Thank you for the rebuttal. I have no further questions.

---

> > > ### Author Response · Authors · 2026-04-04
> > >
> > > We are truly pleased that our responses helped clarify your concerns, and we sincerely appreciate the time and effort you devoted to providing such positive and constructive comments on our work.
> > >
> > > Thank you again for your thoughtful review and consideration.

---

### Official Review · Reviewer_hB48 · 2026-03-09

**Soundness:** 2
**Presentation:** 2
**Significance:** 2
**Originality:** 2
**Overall Recommendation:** 4
**Confidence:** 3

**Summary:**

This paper focuses on human sensing with edge devices that are resource constrained. Human sensing data spans various modalities (IMU, ultrasound, etc.) and data types, and is difficult to collect due to privacy reasons. Annotating such data is also difficult. To address these issues, this paper proposes XTransfer, an approach that adapts pre-trained models and transfers knowledge across modalities using limited data. Specifically, the pretrained models are modified by using a slice-repair-remove pipeline at each layer. Furthermore, a NAS-inspired layer wise search strategy is used to select and recombine layers to reduce the model size. Experimental results are presented on various combinations of source and target datasets to demonstrate the effectiveness of the proposed approach.

**Compliance With Llm Reviewing Policy:**

Affirmed.

**Final Justification:**

The rebuttal addressed my initial concerns and hence I raised my score to weak accept.

**Key Questions For Authors:**

The paper presentation is very dense and is not easy to understand without reading the appendix. Suggest the authors to rewrite by reducing the dependance on appendix in the main paper.

Will the datasets be released? Do authors already have all the approvals for this?

Authors use MetaSense as oracle and mention that it has been trained on extensive target domain data. Is this labeled on unlabeled target domain data?

**Limitations:**

Yes

**Strengths And Weaknesses:**

**Strengths:**
Experimental results show that the proposed approach outperforms various existing transfer learning strategies.

**Weaknesses:**
* Writing
  * Several important details are presented in the Appendix (including layer channel removal which is a core component of the proposed approach), making it a bit difficult to fully understand the paper without reading the appendix. Appendix is supposed to be used for presenting additional information rather than core/important content.
  * The paper emphasizes resource constrained edge devices in the writing but actual evaluation results on edge devices are presented in the Appendix.
* Overall the proposed approach is fairly complex with several steps with numerous components, which could limit its adoption by the community.
* Experimental evaluation uses several private datasets that are not available to others. It is not clear how reproducible these results are and if authors will be releasing these datasets to the community.

---

> ### Author Rebuttal · Authors · 2026-03-28
>
> We thank the reviewer for the thoughtful comments and for recognizing the strong empirical performance of the proposed approach. We agree that the current presentation is dense in places. Because of space constraints in the current submission, some technical details and edge-device results were moved to the Appendix. In the next revision, we will reduce this dependence by bringing clearer explanations of core components (e.g., layer channel removal) and a more visible summary of the edge-device evaluation into the main paper.
>
> **Q1: Will the datasets be released? Do you already have all the approvals for this?**
>
> As stated in the Impact Statement, our sensing data collection was approved by our institution’s Human Research Ethics Committee, all participants provided written informed consent, and the collected data were anonymized and shared only under controlled conditions. We already have part of the approval pathway in place for dataset release, and we commit to obtaining all remaining approvals required for a full release. We will make this commitment more explicit in the paper.
>
> **Q2: How will reproducibility be supported?**
>
> As also stated in the Impact Statement, to support reproducibility, we provide the source code in the supplementary materials and will release the code publicly after the review process, together with detailed instructions to facilitate adoption by the community.
>
> We also provide detailed ablation studies that evaluate each step of the proposed method, which further supports reproducibility and enables the community to verify the role of each component and reproduce the adaptation process.
>
> **Q3: Is the oracle trained on labeled or unlabeled target-domain data?**
>
> The oracle results are based on labeled target data. In Appendix A.1, we state that MetaSense is used as an oracle baseline under the FSL setting, where sufficient labeled target sensing datasets are provided during meta-training.
>
> In addition, following the suggestions of reviewers 1Bf2 and Bxtr, we add new ablation results that provide further empirical support for the effectiveness of the proposed method.

---

> > ### Author Rebuttal · Reviewer_hB48 · 2026-04-01
> >
> > While the rebuttal answers my questions, I still think that the proposed approach is fairly complex. Hopefully the code released by authors will make reproduction easier. So, I am increasing my rating.

---

> > > ### Author Response · Authors · 2026-04-01
> > >
> > > We are truly pleased that our responses helped clarify your concerns, and we sincerely appreciate the time and effort you devoted to providing such positive comments on our work. We will also release the code to fully support adoption by the community.
> > >
> > > Thank you again for your thoughtful review and consideration.

---

### Official Review · Reviewer_1Bf2 · 2026-03-10

**Soundness:** 3
**Presentation:** 3
**Significance:** 3
**Originality:** 3
**Overall Recommendation:** 4
**Confidence:** 2

**Summary:**

The paper presents XTransfer, a modality-agnostic, few-shot model transfer framework designed for human sensing applications on resource-constrained edge devices. The framework addresses the challenges of data scarcity and modality shifts by utilizing a Splice-Repair-Removal (SRR) pipeline to align latent features (via Mean Magnitude of Channels) between pre-trained source models and target sensor data in a reduced PCA space. To accommodate edge deployment, XTransfer incorporates a Layer-Wise Search (LWS) mechanism to selectively recombine and compress the adapted layers from multiple source models. The authors provide extensive evaluations across diverse sensing modalities and demonstrate the framework's effectiveness on real-world edge hardware testbeds (e.g., Raspberry Pi, smartphones, smartwatches).

**Compliance With Llm Reviewing Policy:**

Affirmed.

**Final Justification:**

The author's response has resolved my issue. I hope to maintain my positive recommendation.

**Key Questions For Authors:**

To better isolate the contribution of the SRR adaptation module from the LWS compression module, could you provide a brief ablation (or discussion) where a baseline method (e.g., DAPN) is subjected to a standard post-hoc compression technique (like L2-based pruning)? How does the ATR ratio compare in that controlled setting?

**Limitations:**

Yes.

**Strengths And Weaknesses:**

Strengths:
1.The paper addresses a highly practical bottleneck in ubiquitous computing and edge AI: the prohibitive cost of collecting large-scale sensor data.
2.The authors evaluate the method not only on standard public datasets but also on privately collected real-world datasets encompassing diverse modalities (IMU, ECG, Ultrasound, mmWave). Furthermore, the on-device latency and memory profiling on commercial edge devices strongly substantiate the claims of resource efficiency.
3.Utilizing the Mean Magnitude of Channels (MMC) combined with a PCA subspace for distribution alignment is a lightweight and effective engineering choice that avoids the heavy computational burden of traditional cross-modal semantic alignment.
Weaknesses:
The empirical evaluation fundamentally conflates the benefits of feature adaptation with those of model compression. In Table 4 and Table 12, XTransfer is compared against standard meta-learning or transfer learning baselines (e.g., MAML, DAPN, ProtoNet) that preserve the entire dense ResNet-18 architecture. The vastly superior ATR ratios reported are structurally guaranteed by the pruning/NAS components. To isolate the algorithmic contribution of the proposed adaptation method, it must be compared against baselines equipped with equivalent compression budgets (e.g., applying standard channel pruning or LWS to DAPN/MAML).

---

> ### Author Rebuttal · Authors · 2026-03-28
>
> We thank the reviewer for the positive assessment and for highlighting the practical importance of the problem, the breadth of the cross-modality evaluation, the real-world edge profiling, and the lightweight MMC+PCA design. We also appreciate the key suggestion on isolating the contribution of SRR from LWS. To address this, we add new ablation results with discussion below.
>
> |                |    **3-shot**    |                 |    **5-shot**    |                 |   **10-shot**    |                   |
> |----------------|:----------------:|:---------------:|:----------------:|:---------------:|:----------------:|:-----------------:|
> | **Method** | **Accuracy (\%)** | **ATR** | **Accuracy (\%)** | **ATR** | **Accuracy (\%)** | **ATR** |
> | ProtoNet |     50.6       |       0.51      |      45.7       |       0.46      |      49.8      |       0.50        |
> | DAPN |     48.7       |       0.49      |      51.2       |       0.51      |      54.7      |       0.55        |
> | DAPN+Pruning |     25.0       |       0.66      |      28.3       |       0.64      |      30.0       |       0.77        |
> | MAML |     36.6       |       0.37      |      42.3       |       0.42      |      42.4      |       0.42        |
> | SemiCMT |     33.3       |       0.33      |      38.9       |       0.39      |      48.9      |       0.49        |
> | GPT2 |     41.0       |       0.11      |      34.0       |       0.09      |      45.0      |       0.12        |
> | SRR-w/o-Removal  |      58.0       |       0.27       |      63.6     |       0.30       |      71.2      |      0.33        |
> | SRR |      53.7      |       0.30       |      61.8     |       0.36      |      71.6      |      0.42       |
> | Our-Single |     59.0       |       1.42      |      71.8      |       1.57      |      80.7       |       2.09        |
>
>
> **Q1: How does the suggested baseline with post-hoc compression technique compare under a matched compression budget?**
>
> Since LWS is not directly compatible with the baselines, we apply the L2-based structural pruning (also used in Fig. 1(c) and Appendix C.6), to DAPN. For a fair comparison, we set the pruning rate such that the pruned DAPN matches the average model size of Our-Single under each shot setting. We then report both Accuracy and ATR on HHAR, using the same experimental setting as Our-Single in Sec. 6.3 across the 3-, 5-, and 10-shot settings.
>
> As shown in the table above, DAPN suffers a significant accuracy drop on average across all settings after pruning, while ATR on average increases noticeably. Averaged over the three shot settings, accuracy decreases from 51.5\% to 27.8\%, whereas ATR increases from 0.52 to 0.69. This is consistent with the challenge stated in Sec. 1 and the preliminary observation in Fig. 1(c) that existing compression techniques can easily fail to maintain performance under modality shift and few-shot settings, even if they improve resource efficiency.
>
> **Q2: How is the contribution of SRR isolated from LWS?**
>
> We decouple SRR and LWS, and test SRR only and SRR only without the layer channel removal enabled, under the same experimental setting as Our-Single in Sec. 6.3 on HHAR across the three shot settings.
>
> Compared to the Single-source baselines above, the results provide several insights:
> - Without LWS, both SRR-w/o-Removal and SRR still outperform the baselines in accuracy on average, which confirms that SRR alone makes a key contribution under the cross-modality FSL settings. However, they still remain below the Oracle performance, which highlights the importance of LWS.
>
> - In 3- and 5-shot settings, SRR-w/o-Removal achieves higher accuracy than SRR (with layer channel removal enabled). This suggests that, although the proposed layer channel removal is more effective than standard L2-based pruning (as also shown in Appendix C.6), its effect may be less stable in very low-shot settings when LWS is not enabled. This indicates that the switch control of layer channel removal could be further optimized under different shot settings.
>
> - Both SRR-w/o-Removal and SRR achieve relatively low ATR. This indicates that, although the layer-wise connectors are lightweight, they still introduce additional resource costs and may slow down edge deployment if no further structural optimization is applied. In addition, SRR achieves higher ATR than SRR-w/o-Removal, showing that the proposed layer channel removal is effective in reducing part of model resource overhead.
>
> - After enabling LWS, both accuracy and ATR of Our-Single improve significantly. On average, Our-Single improves accuracy by 6.2\% and 8.1\% over SRR-w/o-Removal and SRR, respectively, while improving ATR by 5.64 times and 4.70 times. This shows that LWS also makes a major contribution by further improving both performance and efficiency on top of SRR.
>
> Overall, we will include the discussion in the next revision of the paper.

---

> > ### Author Rebuttal · Reviewer_1Bf2 · 2026-04-03
> >
> > My question has been answered, and I have no further questions.

---

> > > ### Author Response · Authors · 2026-04-03
> > >
> > > We are truly pleased that our responses helped clarify your concerns, and we sincerely appreciate the time and effort you devoted to providing such positive and constructive comments on our work.
> > >
> > > Thank you again for your thoughtful review and consideration.

---

### Official Review · Reviewer_aqDV · 2026-03-13

**Soundness:** 2
**Presentation:** 1
**Significance:** 2
**Originality:** 3
**Overall Recommendation:** 3
**Confidence:** 3

**Summary:**

The paper proposes a few-shot modality transfer methodology for human sensing tasks from data-rich modalities such as image, text, and audio. The central idea lies in minimizing the shift in hidden representations between source and target modalities, measured by MMC shifts, where the layer-wise modulations are gated by a resource-constrained searching procedure. Experiments show that the proposed method outperform existing baselines in various human sensing tasks.

**Compliance With Llm Reviewing Policy:**

Affirmed.

**Final Justification:**

While I still think the paper would need substantial revision for the presentation issue to be fully resolved, the rebuttal addressed a meaningful portion of my concerns, so raise my score to Weak reject.

**Key Questions For Authors:**

Please refer to the questions **Q1**-**Q9** in the Strengths and Weaknesses section.

**Limitations:**

There is no discussion about limitations in the paper.

**Strengths And Weaknesses:**

The proposed method tackles a challenging problem of few-shot modality transfer to human sensing and achieves state-of-the-art performance throughout various human sensing benchmarks.

However, I am highly concerned about the clarity and readability of the overall writing and equations, making hard to understand the method and underlying intuitions. Due to the poor presentation, it is very hard to precisely assess the paper’s other aspects such as soundness and significance.

Throughout the paper, abstract and unclear terminologies are widely used and many explanations are missing details. Readability of the equations and math notations is also poor, due to the overuse of italic fonts for all alphabets and the overuse of in-line math equations.

- There are so many undefined terminologies:

    - **Q1.** What are the source layers and source classes?

    - **Q2.** How are the InterD and IntraD, and D defined exactly?

    - **Q3.** How Cen() calculates the centroid? Is it the same as Mean()?

    - **Q4.** How SRR exactly defined? What is Y_t? Is Y_t conditioned on SRR or Score?

- There are also many unclear explanations:

    - **Q5.** Is the connector shared by modalities or not?

    - **Q6.** What does the sentence “each connector is fine-tuned by the generative transfer module” mean? Is it pre-trained? Does the generative transfer module act as a training objective?

    - **Q7.** Is the adversarial objective for the generative transfer module same as GAN? if so, how are the generator and discriminator implemented? Which distribution does the model learn? Is the adversarial loss jointly trained with the anchor-based repair loss?

    - Anchor class pairing is also hard to understand. Also, it is unclear whether finding pairST is actually beneficial as there is no ablation study.

        **Q8.** how exactly the pairing shift objective defined?

Finally, it remains unclear that why the proposed method is particularly targeted to transferring to human sensing tasks, as the overall framework seems to be modality-agnostic.

**Q9.** Can the proposed method also applied to different source and target pairs, other than human sensing? If so, could the authors provide the comparisons between baselines?

---

> ### Author Rebuttal · Authors · 2026-03-28
>
> We thank the reviewer for the careful reading and for recognizing the importance, originality, and strong empirical performance of our work. We note that the main concerns are presentation clarity, relating to notation, definitions, and the explanation of the pipeline. We provide concise clarifications below and will improve readability in the next revision.
>
> **Q1.** Source layers refer to the L-units (Sec. 3) segmented from each pre-trained source backbone, where an L-unit can be either a single layer or an inseparable dependent block (e.g., a residual block). Source classes are the original label classes of the pre-trained source model, used for anchor-class selection and pairing during repair.
>
> **Q2 and Q3.** D denotes Euclidean distance in the anchor PCA space. InterD and IntraD denote the corresponding inter-class and intra-class distances. Cen() denotes the centroid of the corresponding projected distribution in the anchor PCA space, computed as the mean vector of the projected samples at each layer.
>
> **Q4.** SRR denotes the Splice–Repair–Removal pipeline (Sec. 1). Splice inserts a trainable connector to make heterogeneous layers shape-compatible, repair aligns sensing features with anchor features, and removal discards low-importance channels after repair. Y_t denotes the few target labels used to compute the post-repair S-score (in Appendix B.2). Y_t is an argument of Score() in the layer-value function. It is also involved in repairing through the anchor-based repair loss in Eq. (2).
>
> **Q5.** The connector is not shared across target modalities. In SRR, each candidate layer has its own trainable connector, which is fine-tuned for the current target setting using few target samples. If transferring to a different target modality, the corresponding layer-wise connector is repaired again for that new target setting.
>
> **Q6.** The generative transfer module denotes the optimization stage in which the connector serves as the generator. It enables the repairing by fine-tuning the connector to transform sensing inputs toward the anchor distribution, as shown in Fig. 3. Each connector is trained during repairing and is not pre-trained. The discriminator operates on the corresponding frozen pre-trained source layer together with its anchor PCA space. The module itself is not the objective, while the anchor-based repair loss is.
>
> **Q7.** The adversarial component is GAN-inspired in mechanism, but it is not a standard GAN objective, since during repairing we optimize only the layer-wise connector rather than jointly training both generator and discriminator (Sec. 4.2). The lightweight connector is implemented using a Pre-header, a Resizer, and one encoder–decoder pair (Sec. 4), while the discriminator operates on each frozen pre-trained source layer. The connector is trained to align the sensing MMC distributions with the corresponding anchor MMC distributions in the anchor PCA space. The optimization objective is the anchor-based repair loss in Eq. (2).
>
> **Q8.** The pairing step identifies the one-to-one source–target class matching that minimizes the sum of centroid distances in the aligned feature space, i.e., the pairing-shift objective. We solve this using a standard linear sum assignment (Hungarian) algorithm, which outputs the pairST. This step also aligns the number of classes between source and target, since the source model must provide at least as many classes as the target task.
>
> The key benefit of the pairing step is to enhance cross-modality similarity, formalized by the objective of minimizing the sum of centroid distances, as suggested by Fig. 1(b) (Image vs. Image-Unpair). Since pairing is part of the feature space alignment stage, Appendix C.1 provides supporting evidence for the effectiveness of the overall alignment design.
>
> **Q9.** We focus on human sensing because this setting most strongly combines the two challenges of severe sensing data scarcity and edge resource constraints, motivating XTransfer. This is the main scope of the paper. XTransfer is modality-agnostic because it does not rely on shared semantic spaces or paired cross-modal data, hence it is in principle applicable to other source–target modality pairs beyond human sensing. As a reference, Table 4 also includes results on an image target dataset (ChestX), suggesting applicability beyond sensing-only targets.
>
> In addition, following the suggestions of reviewers 1Bf2 and Bxtr, we add new ablation results that provide further empirical support for the effectiveness of the proposed method.

---

> > ### Author Rebuttal · Reviewer_aqDV · 2026-04-03
> >
> > The rebuttal was helpful in clarifying many of the definitions and pipeline details that were previously hard to follow, and I now have a much better understanding of the method. While I still think the paper would need substantial revision for the presentation issue to be fully resolved, the rebuttal addressed a meaningful portion of my concerns so I raise my score.

---

> > > ### Author Response · Authors · 2026-04-03
> > >
> > > We are truly pleased that our rebuttal helped clarify your concerns and made the method much clearer, and we sincerely appreciate the time and effort you devoted to revisiting the paper so carefully. We also appreciate your constructive feedback that presentation clarity would still benefit from further improvement, and we will substantially improve the writing, notation, and pipeline explanation in the revision.
> > >
> > > We hope the rebuttal and the new ablation results make the paper’s technical value clearer. Beyond the presentation clarity, we believe the paper offers a meaningful contribution on the challenging problem of cross-modality few-shot transfer for human sensing at the edge, supported by broad experiments and strong empirical performance.
> > >
> > > Thank you again for your thoughtful review and consideration.

---

### Decision · Program_Chairs · 2026-04-30

**Decision:**

Accept (regular)

**Comment:**

This paper proposes XTransfer, a novel and practically relevant framework for modality-agnostic few-shot model transfer in human sensing under edge constraints. Across reviewers, there is strong consensus that the paper addresses an important and timely problem, namely, enabling efficient cross-modality transfer with limited data and resources, and introduces a technically interesting solution combining the SRR (Splice–Repair–Removal) pipeline with layer-wise search for efficient model restructuring. The method is evaluated extensively across diverse sensing modalities and real-world edge devices, demonstrating consistent performance gains over existing baselines and strong improvements in both accuracy and resource efficiency. Reviewers particularly appreciate the breadth of experiments, practical deployment considerations, and the lightweight yet effective design choices (e.g., MMC + PCA alignment).

While reviewers raised concerns regarding presentation clarity, complexity of the pipeline, and fairness of certain comparisons (e.g., disentangling adaptation vs. compression), these issues are largely non-fundamental. Importantly, the authors’ rebuttal successfully addressed the main technical concerns, including additional ablations to isolate contributions and clarifications on reproducibility, dataset release, and methodology. Several reviewers explicitly updated their ratings to weak accept after the rebuttal, indicating that the core technical contributions are sound and meaningful.

Overall, despite some remaining presentation and clarity issues, the paper makes a solid and impactful contribution to few-shot cross-modality transfer and edge AI. The strengths outweigh the weaknesses, and the work is likely to stimulate further research in this area. The paper is recommended for acceptance.